# Detection and analysis of cloud boundary in Xi'an, China employing 35 GHz cloud radar aided by 1064 nm lidar

Yun Yuan , Huige Di *, Yuanyuan Liu ,Tao Yang , Qimeng Li, Qing Yan, Wenhui Xin, Shichun Li, Dengxin Hua*

*School of Mechanical and Precision Instrument Engineering, Xi'an University of Technology, Xi'an 710048, China*

\* Corresponding author: dihuige@xaut.edu.cn, dengxinhua@xaut.edu.cn

Lidar @1064 nm and Ka-band millimeter-wave cloud radar (MMCR) are powerful tools for detecting the height distribution of cloud boundaries, and can monitor the entire life cycle of cloud layers. In this study, lidar and MMCR are employed to jointly detect cloud boundaries under different conditions. By enhancing the echo signal of lidar @1064 nm and combining its Signal-to-noise ratio ($SNR$), the cloud signal can be accurately extracted from the aerosol signals and background noise. The interference signal is eliminated from Doppler spectra of the MMCR by using the noise ratio of the smallest measurable cloud signal ($SNR_{min}$) and the spectral point continuous threshold ($N_{ts}$). Moreover, the quality control of the reflectivity factor of MMCR obtained by the inversion is conducted, which improves the detection accuracy of the cloud signal. We analyzed three typical cases studies; case one presents two interesting phenomena: a) at 19:00–20:00 CST (China standard time), the ice crystal particles at the cloud top boundary are too small to be detected by MMCR, but they are well detected by lidar. b) at 19:00–00:00 CST, the cirrus cloud tranists to altostratus where the cloud particles eventually grow into large sizes, producing precipitation. Further, MMCR has more advantages than lidar in detection the cloud top boundary within this period. Considering the advantages of the two devices, the change characteristics of the cloud boundary in Xi'an from December 2020 to November 2021 were analysed, with MMCR detection data as the main data and lidar data as the assistant data. The seasonal variation characteristics of clouds show that, in most cases, high clouds often occur in summer and autumn, and the low clouds are usually in winter. The normalised cloud cover shows that the maximum and minimum cloud cover occur in summer and winter, respectively. Furthermore, the cloud boundary frequency distribution results for the whole of observation period show that the cloud bottom boundary below 1.5 km is more than 1%, the frequency within the height range of 3.06–3.6 km is approximately 0.38%, and the frequency above 8 km is less than 0.2%. The cloud top boundary frequency distribution exhibits the characteristics of a bimodal distribution. The first narrow peak lies at approximately 1.0–3.1 km, and the second peak appears at 6.4–9.8 km.

**Keywords:** Cloud detection; cloud boundary; Lidar; Ka-band millimeter-wave cloud radar (MMCR); Frequency distribution; Remote sensing and sensors

# 1 Introduction

A cloud is a mixture of water droplets or ice crystals suspended in the air at a certain height through condensation or condensing after the water vapour in the atmosphere reaches saturation (Wang et al., 1998; Zhou et al., 2016; Wild et al., 2012; Stephens et al., 2012). Cloud vertical structure information (Thorsen et al., 2013; Lohmann et al., 2017; Stephens et al., 2005; Wang et al., 1995; Nakajima et al., 1991) reflects the thermodynamic and dynamic processes of the atmosphere and participates in the global water cycle through formation, development, movement,

and dissipation (Wild et al., 2012; Zhang et al., 2012; Zhang et al., 2017; Sherwood et al., 2014; Dong et al., 2010). However, the vertical structure distribution of clouds has great temporal and spatial heterogeneity and a high rate of change, which leads to great challenges in accurately evaluating the radiation effects of clouds at different cloud types and heights. Research on the characteristics of vertical cloud structures has always been an important direction in cloud physics research (Zcab et al., 2019). Cloud boundaries are the main information in the study of vertical cloud structure, mainly referring to the cloud bottom and top boundaries, including the side boundary. The cloud boundary in this study mainly refers to the cloud bottom and top boundaries. Multilayer clouds also include boundary information of intermediate discontinuous clouds (Zhou et al., 2019; Varikoden et al., 2011; Li et al., 2013; Ward et al., 2004; Zhang et al., 2018; Kuji et al., 2013; Kitova et al., 2003; Cao et al., 2021). With the development of remote sensing detection technology, Ka-band millimeter-wave cloud radar (MMCR) (Görsdorf et al., 2015; Kollias et al., 2017; Kollias et al., 2007) and lidar (Apituley et al., 2000; Prot at et al., 2011; Motty et al., 2018; Cordoba et al., 2017) have become effective instruments for cloud boundary detection.

Common methods for detecting cloud boundaries using lidar include the threshold method and differential zero-crossing method. The threshold method (Kovalev et al., 2005) uses a background signal to measure the echo signal amplitude. The first point where the echo signal is higher than the background signal and exceeds the set threshold is the cloud bottom boundary. However, because of the existence of noise, a point with a marked increase in amplitude may not be found under the condition of a low signal-to-noise ratio ($SNR$); therefore, the cloud bottom boundary cannot be judged. Pal et al. (1992) proposed the differential zero-crossing method through Calculation of dP/dr using lidar backscattering intensity $P$ and range $r$, and the first derivative of backscatter intensity $dP/dr$ changes sign from negative to positive and this zero crossing is cloud bottom. The threshold, differential zero-crossing, and variant detection methods are all based on the feature points of cloud boundaries (Streicher et al., 1995). They are easily affected by noise, and some indicators must be introduced in the specific implementation process to determine the cloud boundary by changing the experience threshold frequently during calculation, which causes difficulties in accurate cloud boundary detection. Young et al. (1995) designed an independent double-window algorithm to detect cloud bottom and top boundaries by combining the lidar signal and a known atmospheric backscatter signal. However, the algorithm needs to manually adjust the window size or the selection of the threshold. Based on the wavelet covariance transform method, Morille et al. (2007) determined the local maxima on both sides of the cloud peak as cloud bottom and cloud top, but this method mistake some real signals at the cloud bottom as noise and miss some information at the cloud top, and resulting in overestimation and underestimation of cloud base and cloud top height respectively. Mao (2011) adopted a multiscale hierarchical detection algorithm, selected the starting and ending points of the feature area as the cloud bottom and cloud peak, and detected the cloud top and bottom through multiple iterative updates.

The determination of the cloud boundary by MMCR is mainly based on the threshold of the echo reflectivity factor used to detect the cloud boundary (Hobbs et al., 1985; Platt et al., 1994). Kollias et al. (2007) judge step by step from the bottom to the top of the reflectivity. If the $SNR$ of more than nine consecutive distance gates reaches the set threshold, these gates represented as cloud signals; otherwise, it is deemed a noncloud signal. Clothiaux et al. (1999) used 35 GHz millimeter wave cloud measuring radar to analyse different types of clouds and considered that the dynamic range of the cloud reflectivity factor is from -50 to 20 dBZ. The existence of certain ground object

echoes and biological groups (including insects and other biological particles) in the lower atmosphere interferes with real cloud echo signals (Luke et al., 2008; Görsdorf et al., 2015; Oh et al., 2016; Melnikov et al., 2013; Melnikov et al., 2015). If the subjective reflectivity factor threshold is directly used to determine the cloud signal, it is not suitable for all cloud types. Therefore, when a cloud signal cannot be accurately identified, large errors in the detection of cloud boundaries result.

Research on the macro- and microscopic structures of clouds in a specific area mainly relies on ground-based observations. Currently, for better cloud detection, it is necessary to combine lidar and MMCR to observe and study local clouds (Sauvageot et al., 1996; Intrieri et al., 1993; Wang et al., 2000; Sasse et al., 2001; Borg et al., 2011; Delanoe and Hogan, 2008). This study combined the advantages of lidar and MMCR in detecting clouds to achieve high-precision cloud boundary detection and inversion. We effectively identify cloud signals from Doppler spectra data of MMCR, and through data quality control, the interference signal caused by floating debris is eliminated to improve the detection accuracy of the cloud boundary. Based on the idea that the MMCR only presents the cloud signal to make cloud boundary detection simple and easy to operate, in this study, we effectively separate the cloud signal from aerosol and background noise by enhancing and transforming the lidar signal and combining the SNR (Xie et al., 2017) to realise the accurate detection of cloud boundaries. By analysing the results of cloud boundary detection by the two instruments under special weather conditions in Xi'an, the cloud boundary evaluation criteria for the joint observation of the two instruments are established, and the variation characteristics of cloud boundary height over Xi'an in 2021 are statistically analysed in detail.

## 2 Observation and Instrument

Xi'an City (107°.40'-109°.49'E, 33°.42'-34°.45'N), Shaanxi Province (105°29'-111°15'E, 31°42'-39°35'N) is located in the Guanzhong Basin in the middle of the Weihe River Basin, bordering the Weihe River and Loess Plateau to the north and the Qinling Mountains to the south. Xi'an has a semi-humid climate. Owing to its special geographical location, it is particularly urgent to analyse cloud observations and analyses in Xi'an. The lidar and MMCR are installed at the Jinghe National Meteorological Station in China, placed side-by-side at a distance of 50 m, and both adopt the vertical observation mode to obtain the vertical structure information of clouds. Black line represents Shaanxi Province, dark blue represents the Yellow River, wathet blue represents the Weihe River, and red dot indicates the location of the Jinghe National Meteorological Station in Fig. 1.

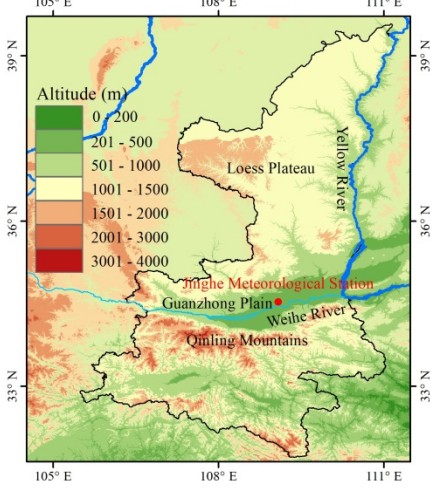

Fig. 1. Geographical coverage of Shaanxi Province (105°29'-111°15'E, 31°42'-39°35'N). The red dot indicates the location of the Jinghe National Meteorological Station in Xi'an (107°.40'-109°.49'E, 33°.42'-34°.45'N).

The lidar used in this study was developed by Xi'an University of Technology. The MMCR is the HT101 all-solid-state cloud radar researched by Xi'an Huateng Microwave Co., Ltd. The main parameters are listed in Tables 1 and 2, respectively.

Table 1 Main parameters of the lidar

| Indicators | Devices | Main parameter |
|---|---|---|
| Launch system | Laser | Nd:YAG; 0.75J@1064nm |
| Receiving system | Cassegrain telescope | Φ400 mm |
| | Filter | 0.5 nm |
| Detection system | Detector | APD |
| | Sampling mode | Analog detection |
| Spatiotemporal resolution | Time resolution | 2 min |
| | Range resolution | 3.75 m |
| Pulse accumulation | | 2000 |

Table 2 Main parameters of MMCR

| Indicators | | Detailed description |
|---|---|---|
| Radar system | | All solid-state; All coherent Doppler; Pulse compression |
| Working frequency | | 35 GHz, and wavelength is 8.6 mm |
| Detection altitude range | | 15 km |
| Detection blind area | | 150 m |
| Spatiotemporal resolution | Time resolution | 5 s |
| | Range resolution | 30 m |
| Scanning mode | | Vertical headspace fixed pointing |
| Pulse width | | $1\ \mu s$、$5\ \mu s$、$20\ \mu s$ |
| Detection accuracy | | $Z{\leq}0.5\ dB$、$V{\leq}0.5\ m/s$、$W{\leq}0.5\ m/s$ |

# 3 Method

Using active instruments to determine cloud boundaries through remote sensing measurements, echo signals in clear-sky areas decay rapidly with increasing detection distance. When a cloud signal is detected, the amplitude of the echo signal begins to increase sharply. Usually, during the actual observation, the background noise or aerosol layer also increases the amplitude of the echo signal, but the backscattering intensity of the cloud layer is more continuous and stronger than the aerosol layer and background noise. Therefore, cloud layer and cloud boundary detection can be realised according to the characteristic changes in the echo signals.

## 3.1 Lidar cloud boundary detection

The lidar equation owing to elastic backscattering (Wandinger, 2005; Motty et al., 2018) can be written as,

$$P(\lambda,r) = P_0 \frac{c\tau}{2} A\eta \frac{O(r)}{r^2} \beta(\lambda,r) \cdot \exp\left[-2\int_0^r \sigma(\lambda,r)dr\right],\tag{1}$$

where λ is the wavelength of the emitted light, $r$ represents the detection distance, and $\beta(\lambda,r)$ and $\sigma(\lambda,r)$ are the atmospheric backscattering and extinction coefficients, respectively. $O(r)$ is the laser-beam receiver field-of-view

overlap function, $c$ is the speed of light, $P_0$ is the average power of a single laser pulse, $\tau$ is the temporal pulse length, $\eta$ is the overall system efficiency, and $A$ is the area of the primary receiver optics responsible for the collection of backscattered light.

Considering the influence of the background noise and response noise of the photomultiplier detector, Eq. (1) can be further expressed as

$$P(\lambda,r) = C \cdot \frac{\Delta r}{r^2} \cdot \beta(\lambda,r) \cdot \exp\left[-2\int_0^r \sigma(\lambda,r)dr\right] + E(\lambda,r) + N_{back}(\lambda,r'),\tag{2}$$

where $C$ is the system constant, which is determined by the laser energy, receiving area of the telescope, and quantum efficiency of the detector. $\Delta r$ is the detection range resolution of the system. $N_{back}(\lambda,r')$ is the background noise received by the system. $E(\lambda,r)$ represents the noise introduced to the detection system by calibration.

To avoid amplifying the high-level noise signals, we do not perform distance square correction in Eq. (2) but directly process it as follows:

$$P_{new}(\lambda,r) = \frac{P(\lambda,r) - E(\lambda,r) - N_{back}(\lambda,r')}{C \cdot \Delta r}.\tag{3}$$

For ground-based lidar, the echo signal at a certain height range (>15 km in this study applied to the Xi'an region) can be considered as molecular scattering, $N_{back}(\lambda,r')$ can be estimated with the signal within this range, and the standard deviation of the noise within the distance range is calculated as follows:

$$Sd = \left[\frac{1}{n-1}\sum_{i=1}^{n}\left(x_i - \frac{1}{n}\sum_{i=1}^{n}x_i\right)\right]^{\frac{1}{2}},\tag{4}$$

where $x$ denotes the background noise signal. The noise of the lidar signal can be expressed as

$$\text{Noise}(r) = k \cdot Sd.\tag{5}$$

After the statistical analysis of the system noise, we set $k = 4$ in this study. The algorithm flow chart of detecting cloud boundary by lidar is shown in Fig. 2. Usually, the moving average of $P_{new}(\lambda,r)$ of lidar echo signal is calculated to reduce the influence of random noise. However, the selection of a sliding window directly affects the signal quality. Therefore, wavelet denoising is used to deal with $P_{new}(\lambda,r)$, select symlets7 wavelet base as the wavelet decomposition basis function, the decomposition layer is 5, and the threshold value is the heursure based heuristic threshold value provided by MATLAB. Compared with the smooth function, wavelet denoising can avoid eliminating cloud signals with steep changes due to too much smoothing. Obtaining cloud boundaries mainly includes three parts. The first part is signal preprocessing. $P_{new\_s}(\lambda,r)$ after wavelet de-noising is discretized based on the estimates of noise, and get useful signals $P_{new\_s1}(\lambda,r)$ and $P_{new\_s2}(\lambda,r)$. The second part is to enhance the signal to make the cloud signal sharper from the background noise and aerosol signal. Average signals $P_{new\_s1}(\lambda,r)$ and $P_{new\_s2}(\lambda,r)$ to obtain $P_{new\_sf}(\lambda,r)$. Ascending arrangement are conducted for $P_{new\_sf}(\lambda,r)$ and the new sequence $R_S$ and the corresponding index $id$ are recorded. The maximum and minimum $R_S$ are denoted as $Ma$ and $Mi$, respectively. By building a new mapping proportion coefficient $Pe(i)$, the enhanced signal $P_{new\_sp}(\lambda,r)$ is obtained. Get $Pnew$-$sp$-$smooth$ after smoothing $P_{new\_sp}(\lambda,r)$. The slope $K_1$ of *baseline-1* obtained from the points (15, V1) and (endpoint, V2) on *Pnew-sp-smooth*, and *baseline-2* got by using $K_1$ and point (starting point, V0) as shown in Fig. 3b) and Fig. 4b).Signals exceeding *baseline-2* are regarded as candidate cloud signals as shown in Fig. 3b) and Fig. 4b). The third part is to extract cloud signal and realize boundary detection by combining the *SNR* of echo signal. By fitting the echo signal slope in the height range of 15–20 km, the slope is used as the slope to distinguish the

cloud and aerosol layers (as shown by the magenta line in Fig. 3b and Fig. 4b). Without considering the bottom echo signal (0–2 km), the amplitude of the echo signal received by the lidar decreased with increasing detection height according to the fitted slope, as shown by the blue line baseline in Figs. 3b) and 4b). When the beam senses the presence of clouds, the amplitude of the echo signal will exceed the blue baseline. The *SNR* of the echo signal is an important parameter for distinguishing the cloud and aerosol layers in the echo signal and calculating the *SNR* of $P_{new\_sf}$ using Eq. (6) (Xie et al., 2017),

$$SNR(r,\lambda) = \frac{N \cdot P(r,\lambda)}{\sqrt{N \cdot P(r,\lambda) + N \cdot P_{back}}}, \tag{6}$$

where $N$ is the pulse accumulation, $P_{back}$ is the solar background noise power, and *SNR* in Shannon formula is the power ratio of signal to noise, which is a dimensionless unit. As shown in Figs. 3c) and 4c), the *SNR* of the cloud layer is higher than that of the aerosol layer and background noise, and the *SNR* in the cloud layer is approximately greater than 5 (obtained based on multi-data statistical analysis in different situations). Combined with the *SNR* threshold, the detected cloud information is shown in Figs. 3d) and 4d).Compared with the traditional method of finding cloud bottom and cloud top from echo signals, this method first accurately extracts cloud signals, and then obtains cloud boundaries (cloud bottom and top). This method greatly reduces the interference caused by noise and aerosol signal.

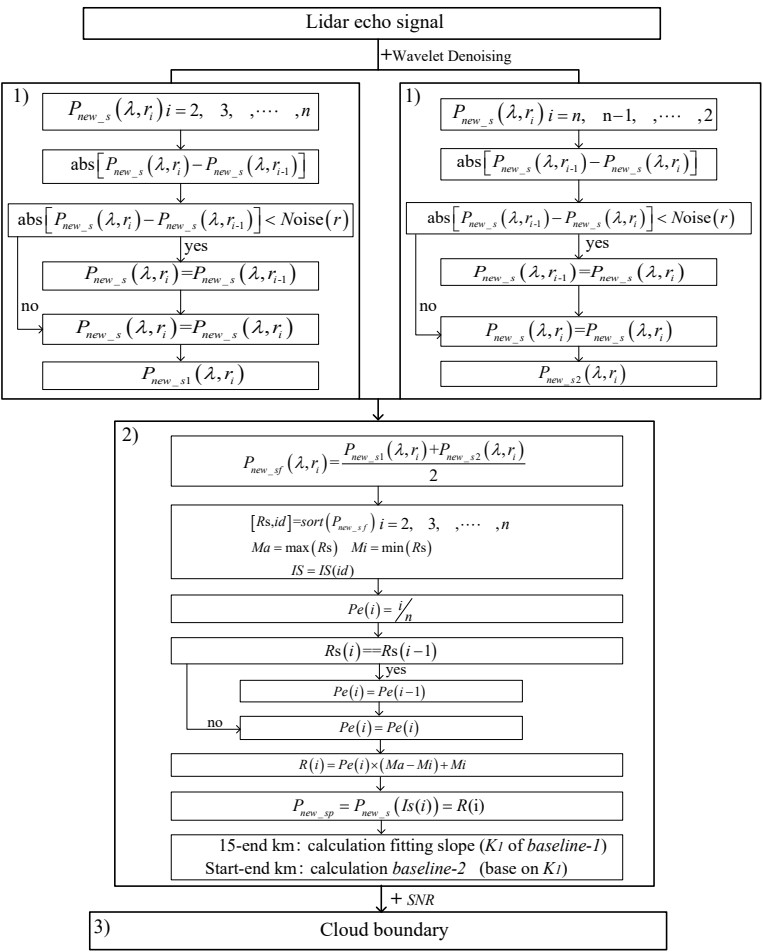

Fig. 2 Use lidar to detect cloud boundary. 1) signal preprocessing, 2) baseline determination based on enhanced signal, 3) identifying cloud boundary with *SNR*

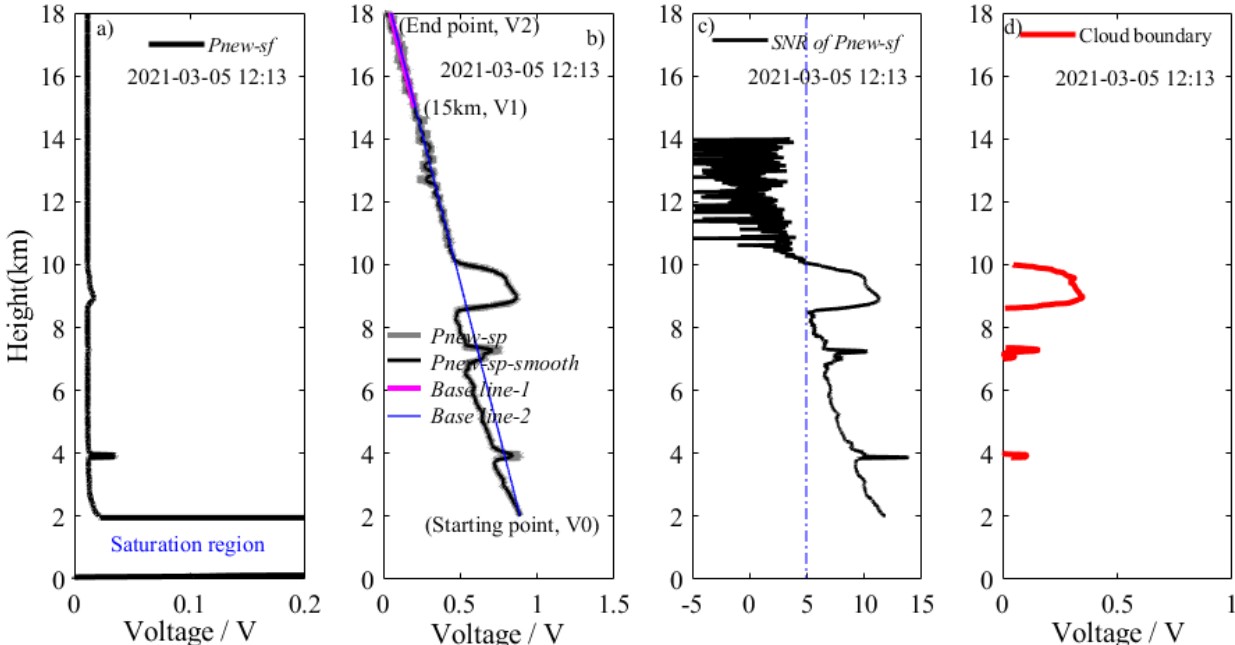

Fig. 3 Detection results of lidar at 12:13 on March 5, 2021. a) $P_{new\_sf}$ of the 1064 nm signal, b) $P_{new\_sp}$ of the 1064 nm signal, c) *SNR* of $P_{new\_sf}$, d) cloud information detected

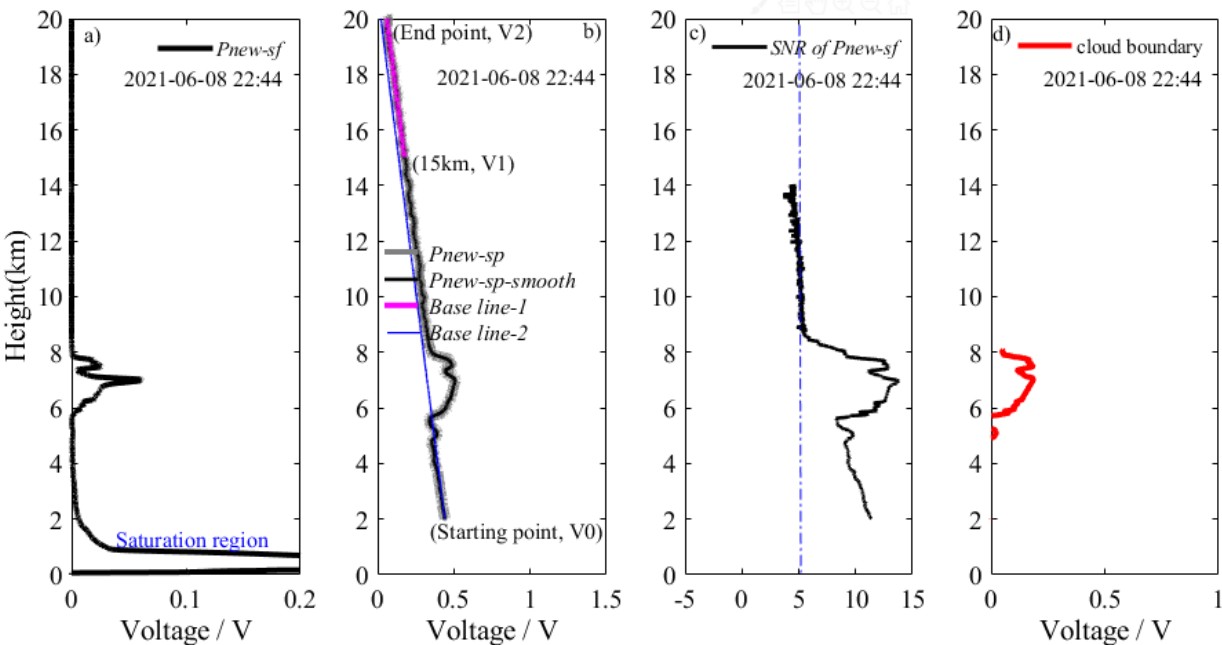

Fig. 4 Detection results of lidar at 22:44 on June 8, 2021. a) $P_{new\_sf}$ of the 1064 nm signal, b) $P_{new\_sp}$ of the 1064 nm signal, c) *SNR* of $P_{new\_sf}$, d) cloud information detected

## 3.2 MMCR cloud boundary detection

Identifying cloud signals from Doppler spectra of the MMCR is affected by the noise level, particularly when the *SNR* is low. As shown in Fig. 5 (Di et al., 2022), if all spectral points above the noise level are integrated, it will result in a large error in the inversion of its characteristic parameters (reflectivity factor, spectral width, radial velocity, etc.). Therefore, it is necessary to carefully identify cloud signals in Doppler spectra signal. Fig. 6 includes two parts: recognition of cloud signals from Doppler spectra of MMCR and data quality control for MMCR. Part

one is mainly to prepare for obtaining effective cloud signals. Generally, cloud signals have a certain number of
continuous spectral points and *SNR*. With the part one of Fig. 6, we use the segmental method to calculate the noise
level, and take it as the noise and signal boundary (as shown is Fig. 5). If spectral data amplitude is greater than
$SNR_{min}$, and search for consecutive velocity bins in its spectral data and record the number of bins. When the
number is larger than $N_{ts}$, and the corresponding spectral signals is determined as an effective spectrum segment.
Intersections of effective spectral segment and noise and signal boundary are left and right endpoints of cloud
spectral, that is, the starting and end point of the spectral moment calculation.
$$SNR_{\min} = \frac{25\sqrt{N_F - 2.1325 + \dfrac{170}{N_P}}}{N_F \cdot N_P} , \tag{7}$$

where $N_F$ is incoherent accumulation, and $N_P$ is the number of fast fourier transform sampling points. The $N_F$ and
$N_P$ of the MMCR used in this study are 32 and 256, respectively, and the $SNR_{min}$ obtained by calculating the $SNR_{min}$
is -17.74 dB. The $SNR_{min}$ is adjusted according to the measured data of the MMCR and $SNR_{min}$ is finally determined
as -20 dB. Based on the research results of Shupe et al. (2008), $N_{ts}$ is set to 7.

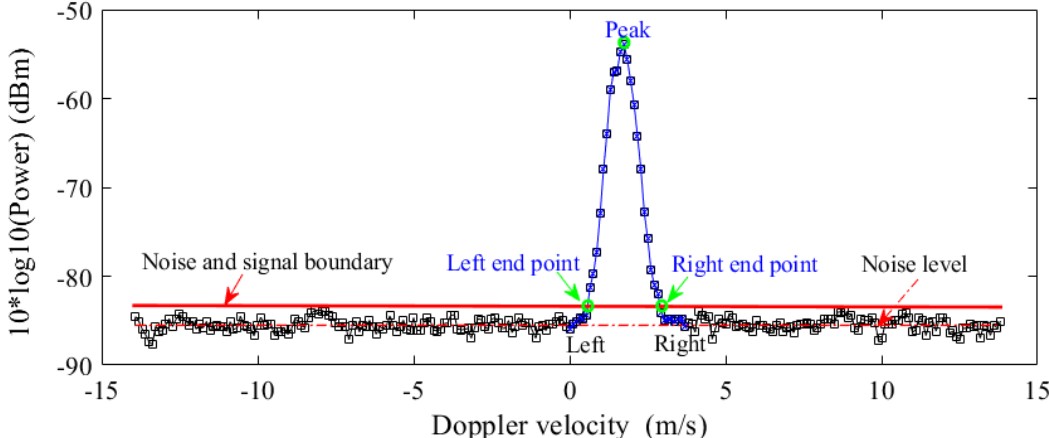

202                            Fig. 5 Schematic diagram of cloud signal recognition in Doppler spectra

The echo signals of floating debris in the low-level atmosphere have the characteristics of a small reflectivity factor,
low velocity, and large spectral width. To further eliminate interfering wave information, we obtained the data
quality control threshold by counting the characteristic changes in planktonic echoes in the boundary layer under
cloud-free conditions. As shown in 2) of Fig. 6, when the reflectivity factor Z <– 20 dBZ, the absolute value of
radial velocity < 0.2 m/s, and the velocity spectrum width > 0.3 m/s are used as the threshold of noncloud
information in bin. If the characteristic parameters of each bin meet the threshold, and assign NaN to the
corresponding bin in reflectivity factor. The echo signals of floating debris in reflectivity factor are eliminated by
the method, and the quality-controlled for reflectivity factor is realised.

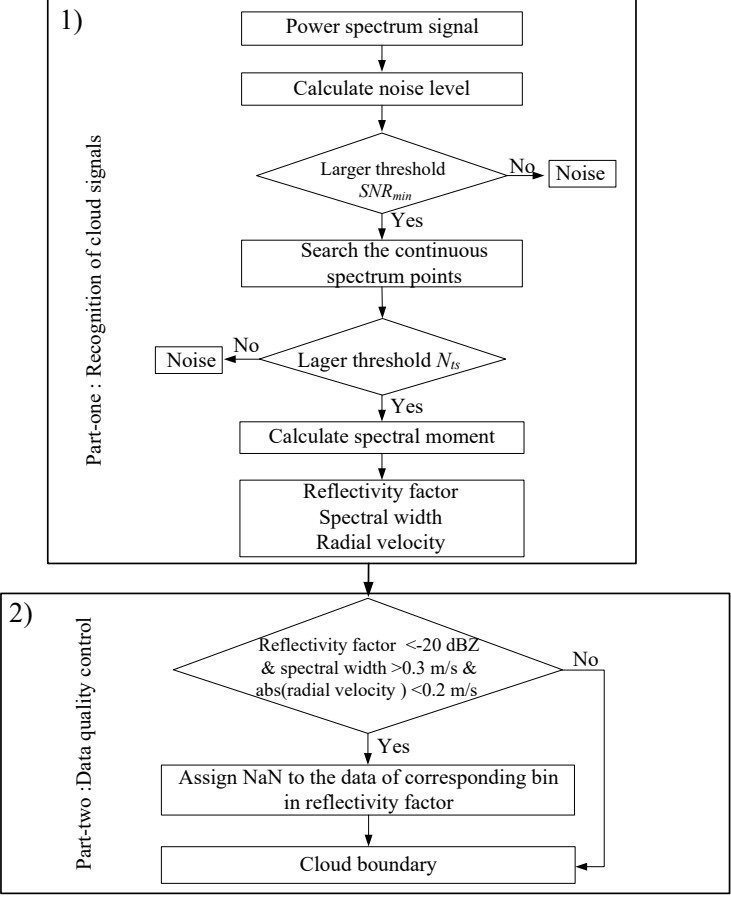

Fig. 6 Flow chart of MMCR cloud boundary detection. 1) recognition of cloud signals from Doppler spectra of MMCR, and 2)
213                                    cloud boundary with data quality control

According to the algorithm flow in Fig. 6, Doppler spectra data at 22:44:00 on 8 June 2021 are analysed to obtain
the cloud signals of the MMCR reflectivity factor, radial velocity, and velocity spectrum width, as shown in Fig.
7a)–c). The noncloud signals at the bottom (0–2 km) are effectively eliminated using the quality control algorithm
shown in 2) of Fig. 6, and the accurate recognition of cloud boundary is realised in Fig. 7d).

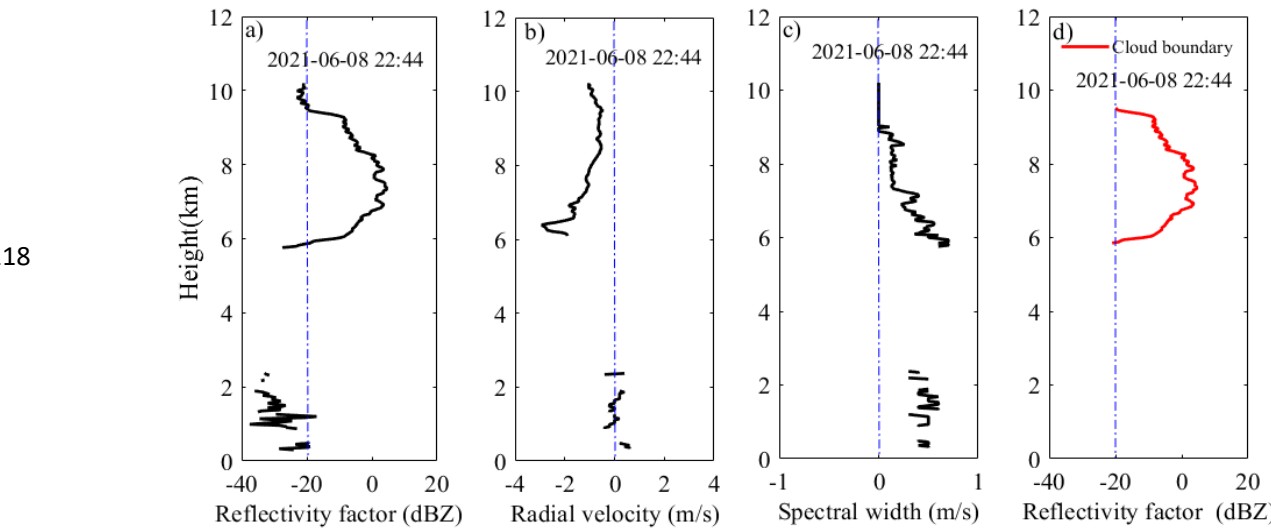

Fig. 7 Meteorological signals of MMCR at 22:44 on 8 June 2021. a) reflectivity factor, b) radial velocity, c) velocity spectrum width, d)
220                                    reflectivity factor after quality control

## 4 Results and discussion

### 4.1 Joint observation and analysis of various types of clouds

Clouds change rapidly (Veselovskii et al., 2017). They often appear in the form of single-layer, multilayer, and precipitating clouds. Section 4 uses the data inversion method proposed in Section 3 to analyse the changing characteristics of clouds under different conditions to obtain reliable cloud macroinformation. Although the spatial and temporal resolutions of the two detection devices are different, their close proximity allows a good 'point-to-point' quantitative comparison between lidar and MMCR. Before data comparison and analysis, the low spatial resolution of MMCR and low temporal resolution of the lidar were interpolated to keep the spatial and temporal resolutions of the two consistent (the time resolution is 5 s, and the spatial resolution is 3.75 m).

1) First case study period

Clouds in the sky often appear as single-layer clouds and the inversion of macroscopic parameters is simpler than that of multilayer clouds. June 08-09, 2021 (19:00–06:00 China standard time (CST)), lidar and MMCR jointly monitored the appearance of monolayer clouds in Xi'an. According to the data method described in Section 3.1, we can obtain cloud change information of time-height-indicator (THI) for $SNR$ of $P_{new\_sf}$ and $P_{new\_sp}$ of lidar @1064nm with a duration of 7 hours, as shown in Figs. 8a) and 8b). The inversion results show that the thickness of the cloud layer is approximately 2 km, and the height of the cloud bottom decreases from 8 km to 4 km with the passage of the observation time. After 05:00 CST, the cloud layer developed deeper, and the laser beam penetrated 0.1 km into the cloud layer and was quickly attenuated. Rainfall begins at 06:00 CST, and the lidar cannot continue effective observation, and the experiment ends. The $SNR$ in Fig. 8a) causes the $SNR$ of the bottom signal to be large (0–2 km, and the echo signal within the range is not considered in the following cases). Cloud signals have a higher SNR than aerosols and background noise. $P_{new\_sp}$ highlights the cloud information from the aerosol signal and background noise, and the details of the instability of the laser energy from 23:00 to 00:30 CST are displayed in Fig. 8b). Combined with the $SNR$ ($SNR > 5.2$ without considering the low-level saturation zone) and $P_{new\_sp}$ thresholds of the cloud signal in Fig. 8a) and 8b), the cloud layer signal detected from the echo signal is shown in Fig. 8c).

Cloud reflectivity factor of the MMCR for the same observation time period, and the cloud signals observed by the two devices have good macrostructural similarity before 06:00 CST. As shown in Fig. 8d), when the quality control of reflectivity factor is not conducted, noncloud signals in the range of 0–2 km are not prominent, and there are some interference signals around the cloud. If we directly detect the cloud boundary with reflectivity factor in Fig. 8a), it will inevitably lead to underestimation or overestimation of the cloud boundary. We can effectively eliminate the noncloud signals at the bottom atmosphere and the interference signals around the clouds using data quality control for the reflectivity factor in Fig. 8e). According to the reflectivity factor of the MMCR, from 03:00 CST to the end of observation, the cloud layer developed deeper, the cloud bottom height gradually decreased from 7 km to 300 m, and the cloud top height developed to ~12 km (the lidar signal fails to show this detail). When rain appeared at 06:00 CST (the microwave radiometer accurately records the rainfall time), MMCR cannot accurately detect the cloud bottom height, but lidar could detect it effectively (the cloud bottom boundary was ~3.8 km). In this case, we can apply lidar and MMCR to detect cloud bottom and top boundaries, respectively, to achieve high-precision detection of cloud boundaries.

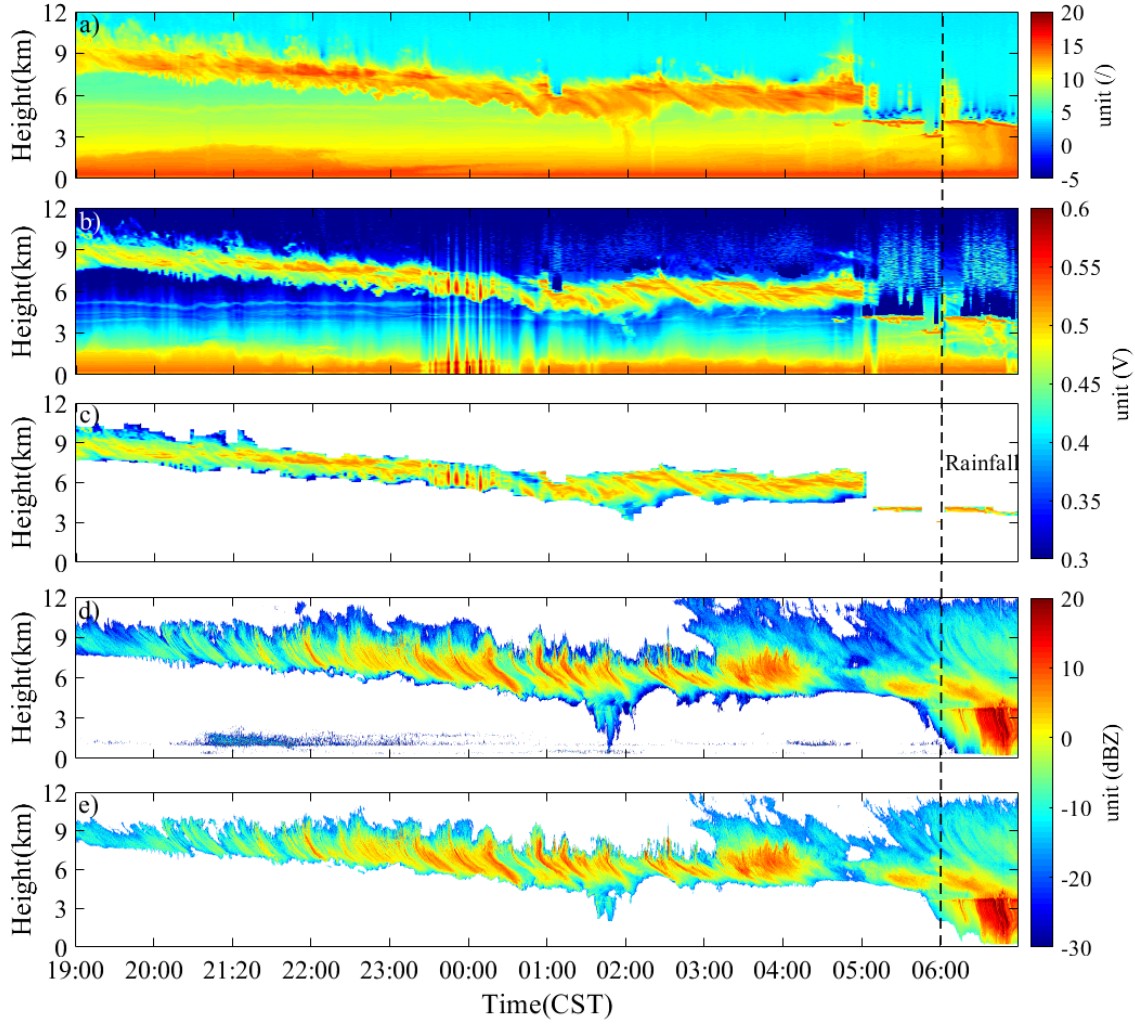

Fig. 8 THI of the echo signal of the lidar @1064 nm from 08 to 09 June 2021. a) *SNR* of $P_{new\_sf}$, b) $P_{new\_sp}$ of the 1064 nm signal, c) cloud information detection results from lidar, d) reflectivity factor without quality control, e) reflectivity factor with quality control (black dotted line indicates rainfall time)

The cloud boundary is retrieved from the cloud signals detected by lidar and MMCR (Fig. 8c and Fig. 8e), and the results are shown in Fig. 9. Between 19:00 and 05:00 CST, the cloud bottom boundary height distributions retrieved by the two instruments were in agreement. Between 21:00 and 06:00 CST, with the development of clouds, the MMCR can detect more cloud information than lidar, especially from 03:00 to 06:00 CST. Although lidar cannot penetrate more clouds during this period, it can provide an effective cloud bottom boundary. At 19:00–20:00 CST, in cloud top boundaries where the ice crystals are too small to be detected by the MMCR, lidar detects the real cloud top. This is attributable to the echo intensity of the MMCR being proportional to the 6th power of the particle diameter, and the lidar echo signal is proportional to the square of the particles. From 19:00 to 00:00 CST, cirrus cloud transition to altostratus, where size of cloud particles increases in the form of collision and finally produces precipitation. In this process, the lidar beam entering the cloud is attenuated, but MMCR has a good advantage in cloud-top detection.

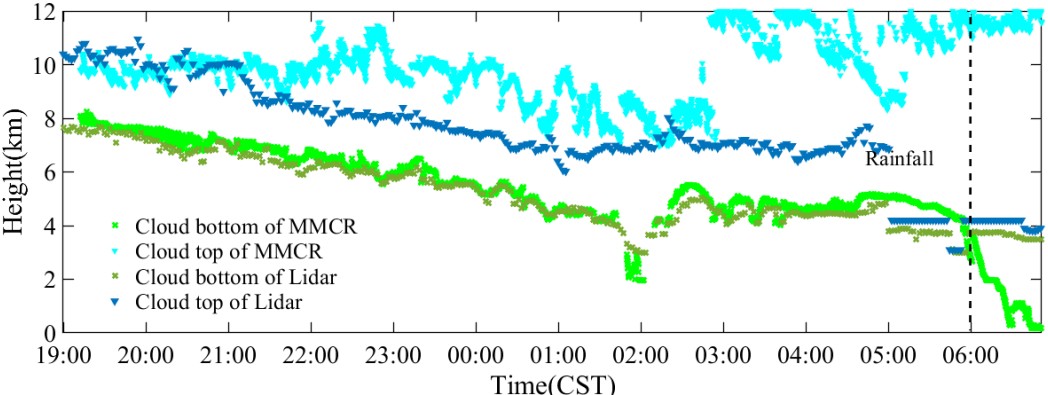

Fig. 9 Cloud boundary detected by lidar and MMCR from 8 to 9 June, 2021

2) Second case study period
From 4 to 5 March 2021, the MMCR and lidar conducted joint observations with a total observation time of 23 h.
By inverting the echo signal of the lidar @1064 nm, we obtained $P_{new\_sp}$ of the echo signal and the $SNR$ of $P_{new\_sf}$,
and the plotted THIs are shown in Figs. 10a) and 10b). These THIs reveal that the double layers of the clouds
appeared in the sky during the observation period. The low-level cloud is located at a height of 4 km, and its
thickness is approximately 2 km; the high-level cloud lies at 7 km, and its thickness is ~2.7 km. The $SNR$ of the
low-level cloud was significantly stronger than that of the high-level cloud, as shown in Fig. 10a). From the
characteristic distribution of the $P_{new\_sp}$ signal in Fig. 10b), the low-level cloud rained from 18:30 to 18:45 CST (the
rainfall time is obtained by checking the microwave radiometer), and the cloud bottom height decreased sharply from
4 km to 0.6 km. Subsequently, the cloud layer gradually dissipated from 2 km to 0.05 km, and the dispersal that
occurred from 02:00 to 10:00 CST was too strong for the lidar to detect more detailed information about the
low-altitude cloud. We also observed the high-level cloud change characteristics shown in Fig. 10b). From 17:00 to
01:00 CST, there was a relatively weak $P_{new\_sp}$ signal in the height range between 7 km and 10 km. This indicates
that the high-level cloud may be in the formation stage at this time, and the particle diameter and number
concentration of clouds are so small that lidar can only receive a very weak echo signal. As the observations
progressed, the development of high-level clouds became relatively mature, and the structure was relatively stable
from 01:00 to 15:00 CST (except at 13:00 CST). Combined with the thresholds of the $SNR$ and intensity
information of the cloud signal in Fig. 10a) and 10b), complete cloud signal detection can be realised, as shown in
Fig. 10c).

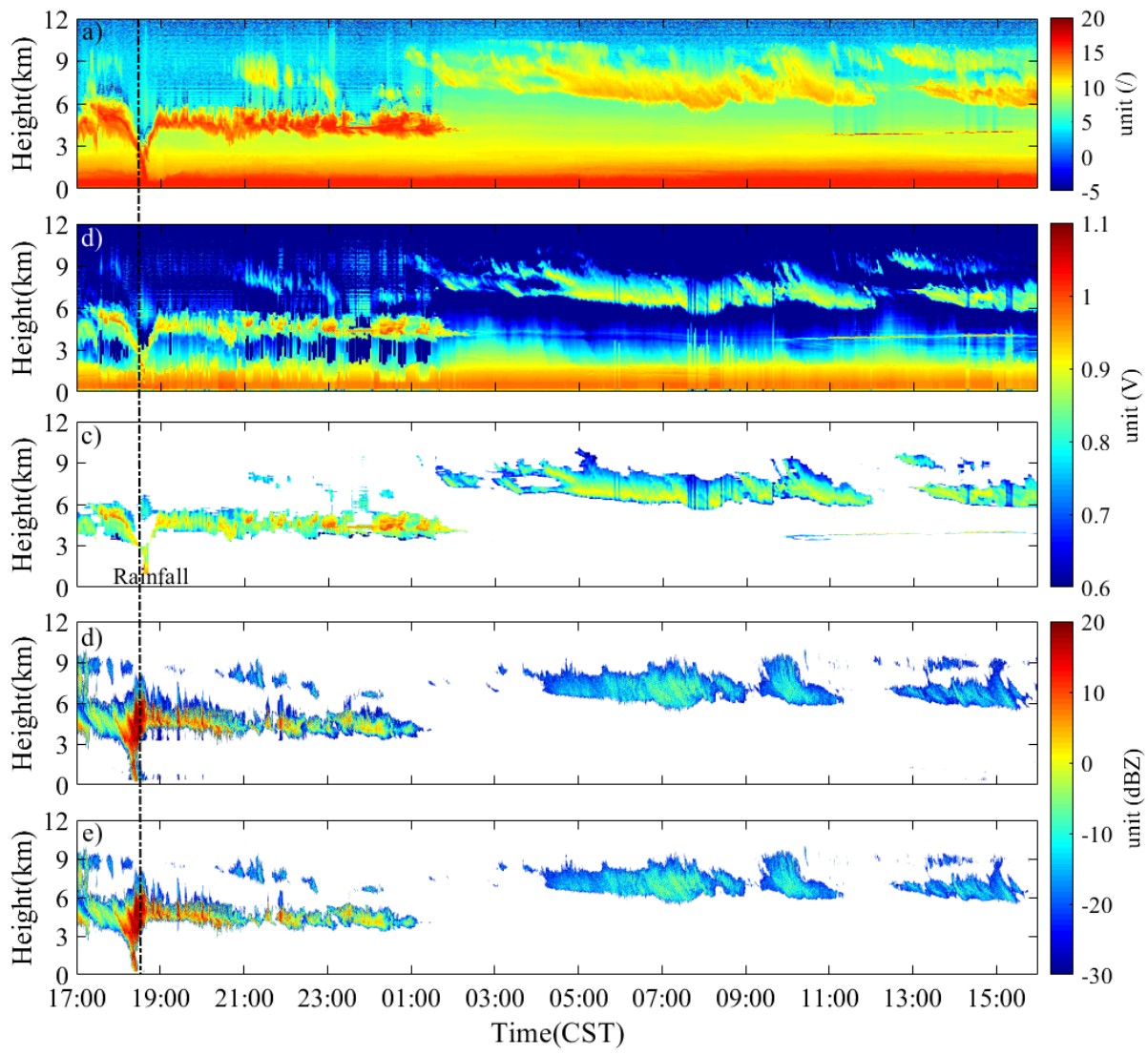


Fig. 10 THI of the echo signal of the lidar @1064 nm from 4 to 5 March, 2021. a) *SNR* of $P_{new\_sf}$, b) $P_{new\_sp}$ of the 1064 nm signal, c) cloud information detection results, d) reflectivity factor without quality control, e) reflectivity factor with quality control (black dotted line indicates rainfall time)

During lidar observations, the MMCR also observed double clouds. Figs. 10d) and 10e) show the signal distribution characteristics of the reflectivity factor of the MMCR without quality control and after quality control, respectively. It can be seen in Fig. 10e) that after data quality control, the noncloud signals and interference signals at the bottom are effectively eliminated. The joint observation results of the lidar and MMCR reveal that the appearance and shape of clouds observed by the two are similar, and the occurrence of rainfall was monitored from 18:30 to 18:45 CST. From 17:00 to 01:00 CST, the penetration ability of the MMCR was markedly better than that of the lidar, and more high-level cloud information was obtained. However, between 01:00 and 04:00 CST for high-level clouds (approximately 8 km), the MMCR detected only part of the debris cloud echo signal, whereas the lidar detected more cloud information. We can speculate that the main reason for this is that clouds were in the growth stage during this time period, their particle diameters were small, or their concentrations were low. The echo signal of the MMCR is proportional to the 6th power of the particle diameter, whereas the echo signal of the lidar is proportional to the 2nd power of the particle diameter; therefore, the lidar can detect clouds that the MMCR cannot

detect. From 10:00 to 15:00 CST, the MMCR also failed to detect the thin cloud signal in the lower layer (a height
of approximately 4 km). Another reason for MMCR failing to detect thin clouds may be that its spatial resolution is
lower than that of lidar, which makes it unable to detect thin clouds.
The height distribution of the double-layer cloud boundaries was detected based on the cloud signals (Fig. 10c and
Fig. 10e) jointly observed by lidar and MMCR, as shown in Fig. 11. The cloud boundary height distribution shows
that the cloud boundary height distributions detected by lidar and MMCR are relatively consistent for low-level
clouds. For high-level clouds, the heights of the cloud bottom boundary detected by the two instruments were
similar, and the cloud top boundary detected by MMCR was higher than that detected by lidar. However, compared
with MMCR, lidar is superior in detecting thin cloud information.

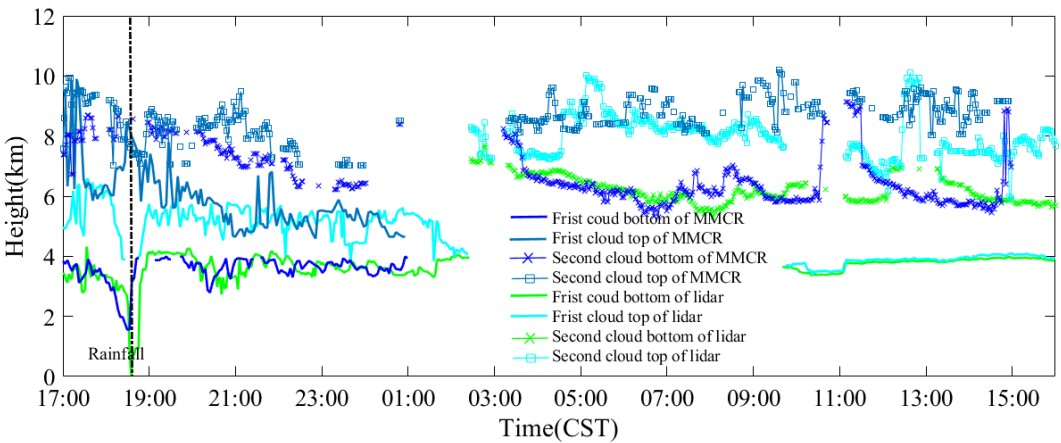


Fig. 11 Cloud boundary detected by the lidar and MMCR from 4 March to 5, 2021
3) Third case study period
On 10 March 2021 lidar and MMCR jointly observed clouds before rainfall for 6 h (06:00–11:00 CST, and began to
rain at 10:45 CST). Fig. 12a) shows the distribution of the *SNR* of $P_{new\_sf}$ with time and space, Fig. 12b) shows the
THI of $P_{new\_sp}$ of the @1064 nm echo signal, and Fig. 12c) shows the cloud signal detected by the thresholds of the
SNR and $P_{new\_sp}$. We inverted the reflectivity factor of the MMCR and performed data quality control operations on
them. The results are shown in Fig. 12d) and Fig. 12e), which are the reflectivity factor of the MMCR without
quality control and quality control, respectively. From the comparison, it is evident that data quality control can
eliminate the interference signal very well, which simplifies the process of merging the high-level convective cloud
and the low-level stratiform cloud.
By comparing the cloud information detected by the lidar and MMCR ( Fig. 12c and Fig. 12e), we can see that
during the period from 06:00 to 10:00 CST, the energy of the lidar beam is severely attenuated at a height of
approximately 4 km, resulting in a very weak echo signal and *SNR* above 4 km. As the observation time progressed,
the phenomenon of virga (> –15 dBZ) occurred in the cloud (Ellis et al., 2011; Williams et al., 2014). The severe
attenuation of lidar in the cloud leads to a sharp decrease in its detection ability, whereas the millimeter wave still
has a strong penetrating ability. When rainfall occurs (the microwave radiometer showed that rainfall occurred at
10:45 CST), neither lidar nor MMCR can effectively identify the cloud bottom boundary, but MMCR can still
detect cloud top boundary information. The height distributions of the cloud boundaries detected by lidar and
MMCR are shown in Fig. 13. The height distribution of the cloud bottom and cloud top boundaries detected by the

two instruments is almost the same from 06:00 to 09:00 CST (the cloud bottom boundary is approximately 3 km, and the cloud top boundary is approximately 4.1 km). A drizzle fell from 09:00 to 10:45 CST, and the lidar obtained an effective cloud bottom boundary. The boundary of the high-level convective cloud at ~8 km and the deep cloud layer from 10:45 CST to the end of the observation period can only be detected by MMCR.

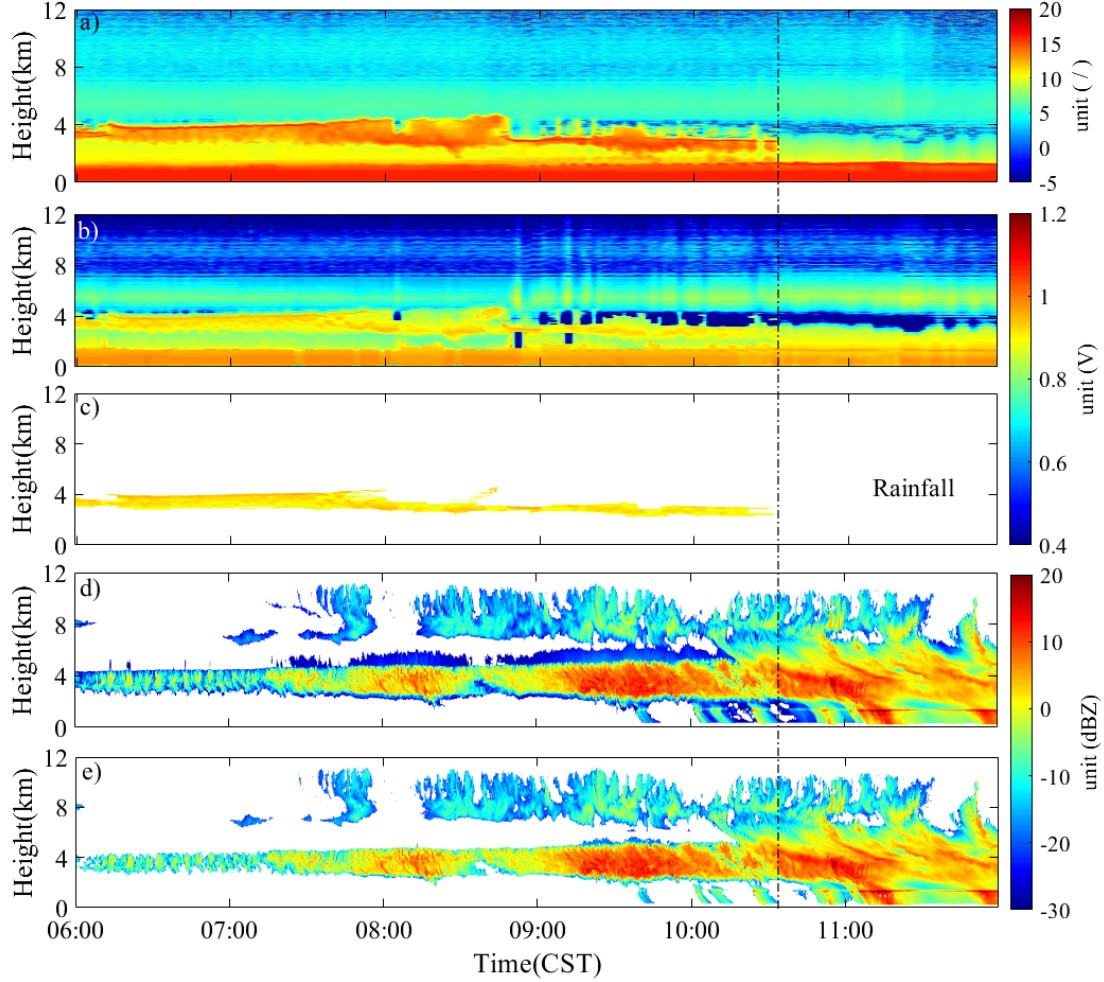

Fig. 12 THI of echo signal of the lidar and MMCR on 10 March, 2021. a) *SNR* of $P_{new\_sf}$, b) $P_{new\_sp}$ of the 1064 nm signal, c) cloud information detection results, d) reflectivity factor without quality control, e) reflectivity factor with quality control (black dotted line indicates rainfall time)

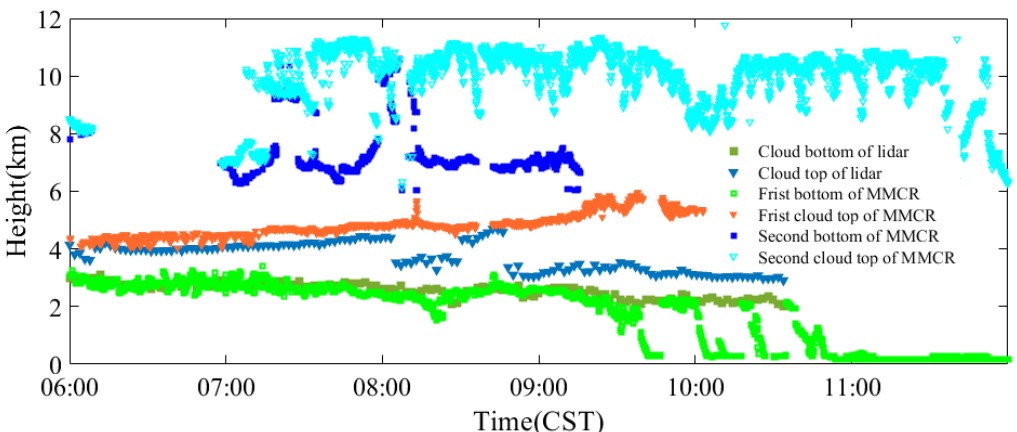

Fig. 13 Cloud boundary detected by the lidar and MMCR on 10 March, 2021

From the differences in the height distribution of the cloud boundaries reached by the two devices in the above three different situations, it can be seen that when a single layer of stratiform clouds appears in the sky, the heights of the cloud bottom boundary detected by the MMCR and lidar are approximately the same. When there are multilayer clouds, MMCR and lidar have good consistency in the detection results of the cloud bottom boundary height of the low-level cloud; however, the energy of the lidar beam attenuates significantly in the low-level cloud, resulting in an inability to fully obtain the effective bottom boundary of low-level clouds and the height boundary of high-level clouds. In this case, the MMCR can obtain more complete height information for the multilayer cloud boundary. Usually, the closer rainfall is, the deeper the cloud layer develops, the more severely the beam of the lidar will be attenuated, and more cloud information cannot be obtained. In other words, MMCR still has the ability to penetrate the cloud layer and detect complete cloud information. Therefore, the joint observation of lidar and MMCR can comprehensively identify and detect cloud boundary conditions in detail. The difference between the cloud boundaries detected by the two may also be due to the different scattering mechanisms of cloud particles to millimeter-wave electromagnetic waves and laser beams or the difference in the methods used by the two devices to determine the cloud boundary; thus, there are some differences in the cloud boundary height results.

## 4.2 Analysis of cloud boundary distribution characteristics in Xi'an

To further analyse the changes in the height distribution of cloud boundaries in Xi'an, we plan to use MMCR and lidar data for cloud boundary analysis. Accordingly, it is necessary to analyse the correlation of the cloud bottom boundary height detected by the two devices. We randomly selected 80 h of data in the joint observation period (to avoid the rainfall period) and calculated the cloud boundary detection results of lidar and MMCR according to the data processing methods in Sections 3.1 and 3.2. As shown in Fig. 14, when the quality control of the MMCR is performed, the correlation between the detected cloud boundary and lidar detection result increases from 0.627 (in Fig. 14a)) to 0.803 (in Fig. 14b)). Moreover, under the premise that the difference in cloud boundaries caused by the different detection principles and algorithms of the two devices cannot be avoided, we can use the cloud boundary data detected by MMCR to replace the missing lidar data.

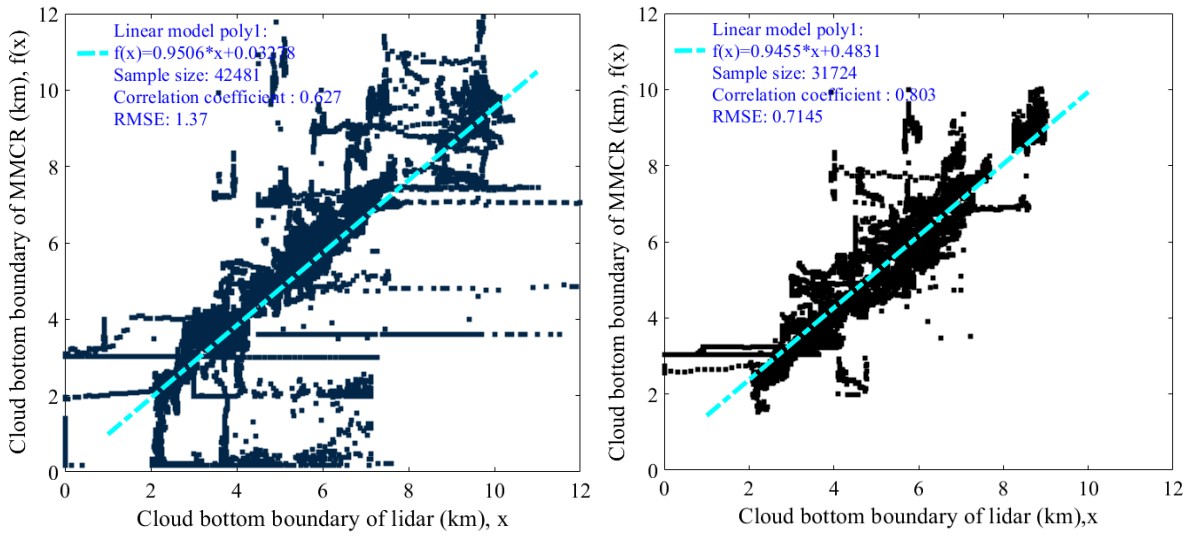

Fig. 14 Correlation between lidar and MMCR cloud bottom. a) without quality control; b) with quality control)

From the above three cloud observation cases, it can be seen that MMCR has more advantages than lidar in detecting cloud-top boundaries. Therefore, when calculating the cloud boundary height distribution characteristics over Xi'an, we only counted the cloud top boundary height detected by the MMCR and considered it as the actual

cloud top boundary. From December 2020 to November 2021, MMCR and lidar stored 302 d (7248 h) and 126 d (872.5 h) of observational data, respectively. During the 12-month observation period, the maximum detection altitude of the MMCR changed. From December 2020 to June 2021, the maximum detection range of MMCR is 12.6 km, and the maximum detection height is changed to 18 km. The total observation hours of MMCR and lidar for each month are shown in Fig. 15. The hours of lidar, MMCR, and simultaneous measurements are 872.5 h. In this study, the four seasons were defined as follows: spring from March to May (MAM), summer from June to August (JJA), autumn from September to November (SON), and winter from December to February (DJF).

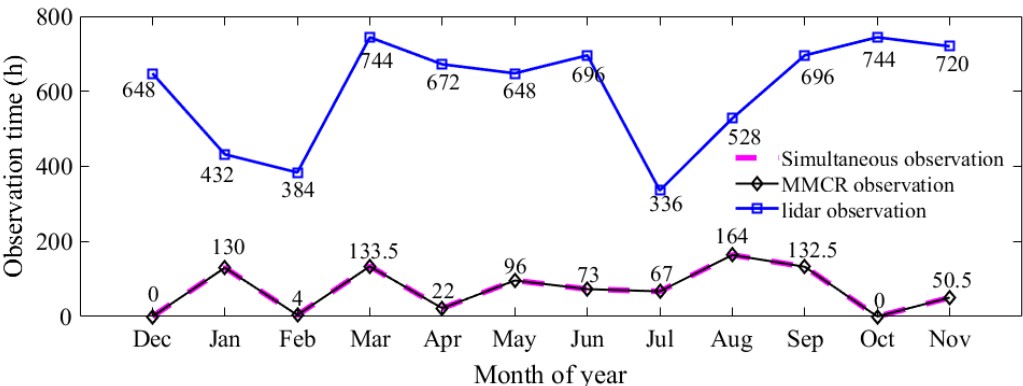

Fig. 15 Monthly observation hours of lidar and MMCR

Table 3 establishes the rules for recording effective cloud bottom information in the observation process using MMCR and lidar under different conditions to improve the detection accuracy of the cloud-bottom boundary.

Table 3 Cloud bottom height recording guideline

| Detection equipment | Observation | Detection conditions | Record cloud bottom boundary |
|---|---|---|---|
| Both the lidar and MMCR | Case 1 | Geometrical thin cloud: the lidar detects bottom; MMCR did not detect the cloud bottom | Results of the lidar |
| | Case 2 | Drizzle: the lidar detects bottom; bottom of MMCR is invalid | Results of the lidar |
| | Case 3 | Both the lidar and MMCR detect cloud bottom | Record the lower value of the cloud bottom boundary |
| MMCR | Case 4 | MMCR detected cloud bottom | Results of MMCR |
| | Case 5 | Drizzle: bottom of MMCR is invalid | No results are recorded |

This study defines 'cloud occurrence frequency' as the ratio of cloud occurrence times to total detection times during the analysed period. The total sample size is $N$, and the sample size of cloud boundaries appearing at different height levels (altitude range from 1.5 km to 12 km is divided into 50 levels) is $n_i$. The seasonal distribution characteristics of the cloud boundary height are calculated according to Eq. (8),

$$y_{\_cloud} = \frac{n_i}{N}(n_i \in N, i = 1....50).$$ 
(8)

Fig. 16 shows the vertical frequency distribution of the cloud boundary seasonally from December 2020 to November 2021. For the vertical distribution of cloud base, the first narrow peaks is the boundary layer clouds ($\leq$ 1.5 km), the second peak is 2.5–3.5 km, and the third peak has a big range in vertical height, which is 4.7–10 km a in spring. Fig.16 (b) shows that the cloud bottom height in summer is mainly distributed at 3–9.5 km, indicating that middle and high clouds may be dominant. The distribution of cloud bottom is bimodal, the first peak is the boundary layer cloud peak, and the second peak is located at 2.7–3.7 km and 3.6–8.3 km in autumn and winter, respectively. The variation in cloud top with seasons shows a bimodal distribution, and spring and summer have a similar trend of cloud top boundary height distribution. The frequency of the cloud top boundary above 10 km was the highest, and the frequency below 2 km was the lowest in summer. The distribution characteristics of cloud top height in autumn and winter indicate that the frequency of low clouds is higher than that in the other two seasons. This is consistent with the results of Zhao et al. (2014) for the SGP site and Xie et al. (2017) for the SACOL site. Although there were some differences in the cloud boundary frequency distribution at some heights, the overall change trend was roughly the same.

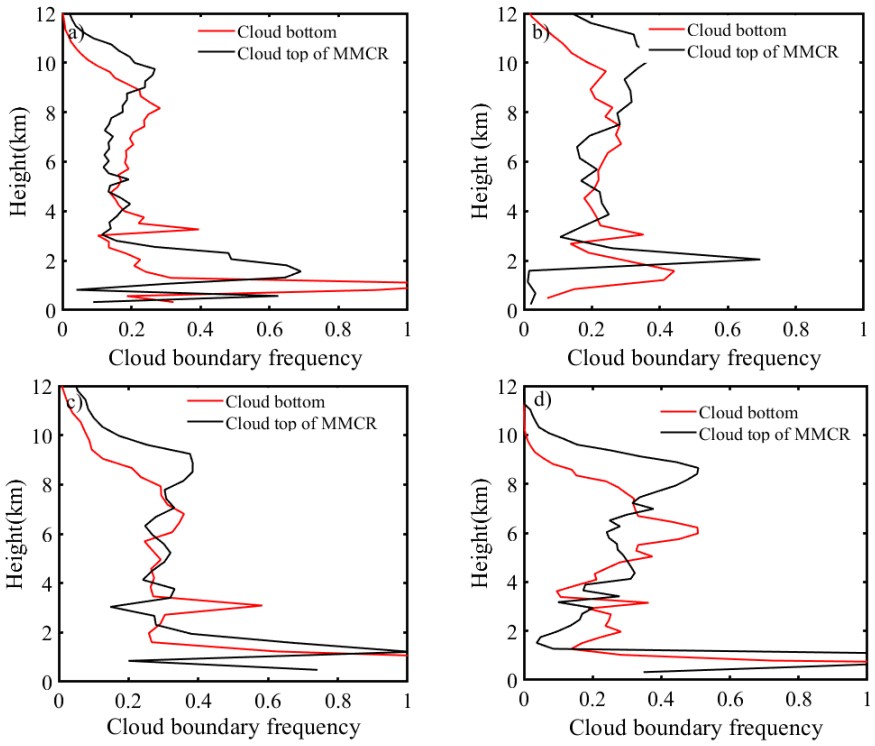

Fig. 16 Frequency distribution of cloud boundaries during (a) spring, (b) summer, (c) autumn, and (d) winter from December 2020 to November 2021 at Xi'an Jinghe National Meteorological Station

Fig. 17 a) shows the monthly variation frequency distribution of clouds. The months with the largest and smallest cloud occurrence frequencies are August and February, respectively. Almost more than 34% of the clouds appear in the form of single layer clouds every month. Compared with January, February, November, and December, the frequencies of double-layer clouds, triple-layer clouds, and more clouds in other months are higher. To show the relative change trend of cloud cover, we calculated the total cloud cover of each month by using the total cloud cover at each time stored by the MMCR. It was found that the maximum cloud cover was in April. Therefore, the

total cover of April was set to 1, and the normalized cloud cover distribution of 12 months was obtained, as shown in the Fig. 17 b). It can be seen from the distribution of cloud cover in every month that the cloud cover is high in summer and the least in winter, indicating that warm atmospheric conditions are more conducive to the formation and development of clouds.

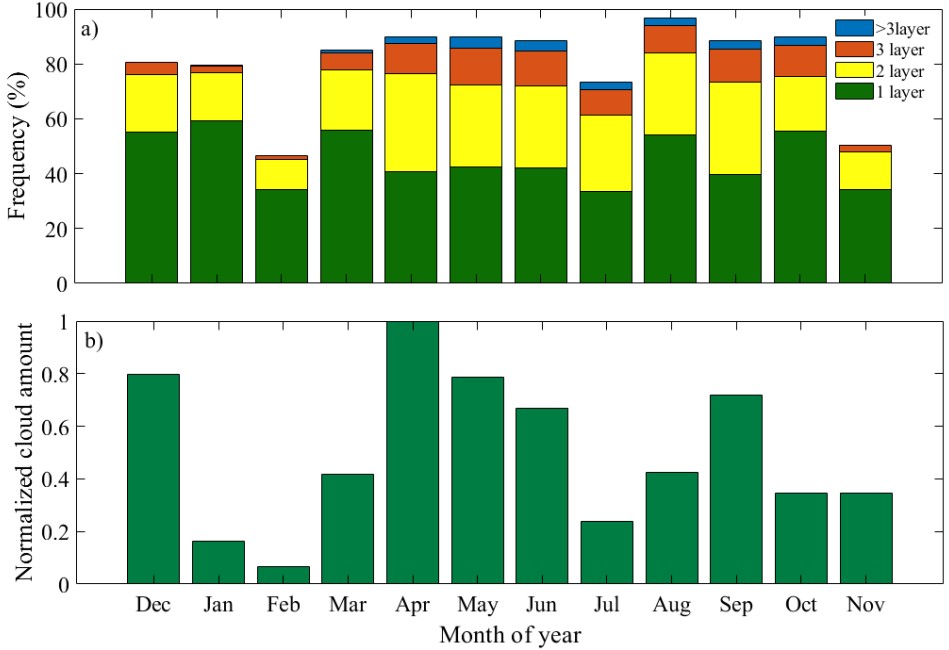

Fig. 17 Monthly variation in cloud frequency distribution and cloud cover from December 2020 to November 2021 a) monthly variation in the frequency of the number of cloud layers. b) monthly variation in cloud cover

As Fig.18 caption says it is the frequency distribution of cloud boundaries observed over Xi'an from December 2020 to November 2021. Frequency of the cloud bottom boundary below the vertical height of 1.5 km is the highest, the frequency within the height range of 3.06 km and 3.6 km is approximately 0.4%, and the frequency above 8 km is less than 0.2%. The frequency of the cloud top boundary at vertical heights has a bimodal distribution, and the first narrow peak is located at 1.0–3.1 km, and the second peak lies at 6.4–10.5 km. Combined with the changing characteristics of cloud layers, it can be seen that during observation in Xi'an, the frequency of clouds below 3.5 km is the largest, and the frequency of high-level ice clouds or cirrus clouds above 8 km is small, which may be due to the limited detection sensitivity of MMCR at the top of high-level clouds where the particles size are very small.

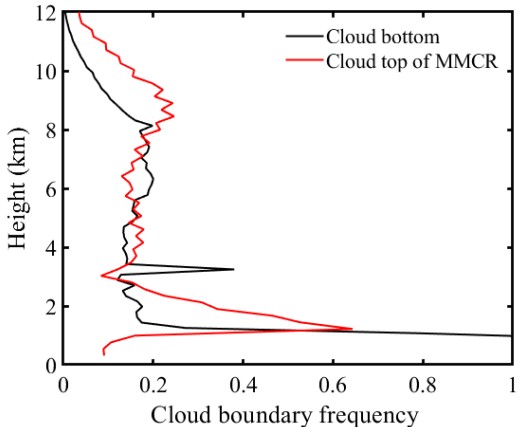

Fig. 18 Frequency distribution of cloud boundaries at vertical heights at Xi'an Jinghe National Meteorological Station from December 2020 to November 2021

## 5 Conclusions

Based on the observation data of lidar, a new algorithm is proposed which can effectively extract cloud signals. Compared with the previous method of identifying cloud bottom and cloud top from echo signals, the new method mainly obtains effective cloud signals through suppressing noise signals and enhancing effective signals to realize cloud boundaries. The algorithm has two main characteristics: 1) in the signal preprocessing, wavelet transform is used for the original signal to avoid the defect of effective information loss caused by improper selection of smooth window; 2) The *SNR* of the signal is considered.

The cloud signals in Doppler spectra are effectively extracted by analyzing the noise level, $SNR_{min}$, and the continuous spectral points of Doppler spectra. The data quality control conditions for MMCR (reflectivity factor < -20 dBZ, spectrum width > 0.3 m/s and radial velocity < 0.2 m/s) were established by analyzing the characteristic of the interference of floating debris signals. By analysing the correlation of cloud bottom height between MMCR and lidar, and the cloud bottom height detection by MMCR with data quality control have a good agreement with lidar (the correlation coefficient is 0.803). Therefore, quality control is an important factor to improve signal accuracy of MMCR.

In this study, combined with the respective advantages of MMCR and lidar in cloud detection, the cloud cover and distribution of cloud boundaries characteristics are analyzed based on the observation data in Xi'an from December 2020 to November 2021.The result reveals that more than 34% of the clouds appear in the form of a single layer every month. The cloud cover was lowest in spring and highest in summer. The seasonal variation in cloud boundary height showed that the distribution characteristics of cloud boundaries in spring and summer were similar, and the frequency of high-level clouds in the range of 8–10 km was greater than autumn and winter. The stratiform clouds appearing below 3.5 km in autumn have the highest frequency, and high-level ice clouds or cirrus clouds above 8 km in winter are less likely to appear. The findings can provide a preliminary analysis of cloud boundary changes in Xi'an. If there are huge amounts of simultaneous observation data of the lidar and MMCR, the comprehensive statistics and analysis of cloud macro and micro parameters in Xi'an can be realized, which can provide better support for the study of climate change characteristics in Xi'an.

## Data availability

The data and code related to this article are available upon request from the corresponding author.

## Author contributions

Conceptualization: Yun Yuan

Investigation: Yun Yuan

Methodology: Yun Yuan and Huige Di

Software: Yun Yuan

Supervision: Huige Di and Dengxin Hua

Methodology and software improvement: Yuanyuan Liu, Tao Yang, Qimeng Li, Qing Yan, Wenhui Xin, and Shichun Li.

Writing – original draft: Yun Yuan

Writing – review & editing: Yun Yuan and Huige Di

Project administration: Dengxin Hua

## Competing interests

The authors declare that they have no conflicts of interest related to this work.

## Financial support

This research has been supported by the National Natural Science Foundation of China, Innovative Research Group
Project of the National Natural Science Foundation of China (grant nos. 42130612, 41627807 and 61875162) and
the Ph.D. Innovation fund projects of Xi'an University of Technology (Fund No.310-252072106).

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
