# Peer review of "Lidar and MMCR applied for the study on cloud boundary detection and the statistical analysis of cloud distribution in Xi'an region"

_Atmospheric Measurement Techniques, 2022_

## Author Response (AR1)

**Response to Editor**

**Dear Editor & Prof.**

We greatly thank you and the two reviewers for the thorough and valuable suggestions to our work, and thank you valuable comments and suggestions by peer experts in the open discussion. The manuscript has been polished and modified by professional organizations, and English has been greatly improved. We have made a point-to-point response to these opinions and suggestions, and believe that the quality of the manuscript has been promoted now. All the comments have been addressed in the revised manuscript, and the responses to each comment are given below.

Thank you very much for considering our work!

Yours sincerely,

Yun Yuan and co-authors
Xi'an University of Technology
yunyuan_91@163.com
dihuige@xaut.edu.cn

**Response to reviewer #1 (in open discussion)**

**General comments:**
This manuscript presents an observational combined lidar (1064 nm) and radar (8.6 mm) data set to determine the cloud boundaries over the ground station located at the Xian region. The authors use signal enhancement techniques to avoid background and aerosol signal thereby improving the SNR for the identification of cloud top and base boundaries. Analysis of one-year data set over the Xian is presented characterizing the cloud cover and single/multiple cloud layer occurrences. Overall, this is an interesting manuscript and has the potential to be published.

We appreciate the reviewer's thoughtful review and constructive comments, which have greatly helped to enrich the details and improve the quality of the paper. These comments have been revised and supplemented in the manuscript, and the responses to each comment are given below. The manuscript has been polished and modified by professional organizations, and I believe that English has been greatly improved.

1. It is well established that combined lidar and radar measurements are essential to monitor the local cloud cycle. This is also acknowledged by the authors here in the manuscript with some references (Line 76) and demonstrated with few case studies (section 4.1). However, the one-year data presented here do not have simultaneous measurements from lidar and radar for about one-third of the total time (7248 hours). This could introduce bias: (a) in the cloud base boundaries as also shown in fig 10 (19:00 – 00:00 hrs) for the cases of cirrus to altostratus transition where the cloud particles would eventually grow into large sizes producing precipitation, and (b) in cloud top boundaries where the ice crystals are too small to be detected by the radar (fig 10, 19:00 – 20:00 hrs). This bias should be mentioned in the abstract and needs to be discussed in the main text.

**Response:** As discussed and analyzed in Section 4.1, MMCR can effectively measure cloud tops compared, but it has no advantage over lidar in detecting cloud bottom. We analyzed the correlation of cloud bottom (0.803) between obtained by MMCR after data control and detected by lidar, and considered that the two instruments have a high correlation for the detection of cloud bottom height. Therefore, we used MMCR data (7248 hours) aided by lidar data (872.5 hours) to improve the accuracy of cloud bottom detection. To reduce the error caused by directly employing MMCR to analyze the cloud bottom. Two biases (a) and (b) have been added to the abstract and discussed in the main text.

Specific modifications are as follows:
1) L 12-L 17: We analyzed three typical cases (e.g., single-layer clouds, multilayer clouds, and precipitating clouds), case one presents two interesting phenomena: a) at 19:00~20:00, the ice crystal particles at the cloud top boundary are too small to be detected by MMCR, which is well detected by lidar. b) at 19:00~00:00, the cirrus cloud transits to the altostratus where the cloud particles would eventually grow into large sizes producing precipitation,and MMCR has more advantages than lidar in detecting the cloud top boundary within this period.

2) L 248-L 254: At 19:00 ~ 20:00, in cloud top boundaries where the ice crystals are too small to be detected by the MMCR, but the lidar detects the real cloud top. The main reason is that the echo intensity of MMCR is proportional to the 6th power of particle diameter, and the lidar echo signal is proportional to the square of particles. From 19:00 to 00:00, the cirrus cloud transits to the altostratus cloud, where the cloud particle size increases in the form of collision and finally produce precipitation. In this process, the lidar beam entering the cloud is attenuated, but MMCR has a good advantage in cloud top detection.

2. I suggest including the lidar wavelength or spectral region in the title (and abstract) of the manuscript, since this often gives the impression that lidar is operated at visible channel (532 nm) – if specific wavelength or type of instrument is not mentioned. Further the term 'statistical analysis' in the title is misleading. To my understanding there is no statistical analysis in this manuscript, rather the authors just present the frequency of cloud top/base altitude occurrences and its seasonal variability.

**Response:** We changed the title of the manuscript to "Detection and analysis of cloud boundary in Xi'an, China employing 35 GHz cloud radar aided by1064nm lidar"

3. Extensive editing of the manuscript is required for the proper English usage.

**Response:** The manuscript has been polished and revised by professional institutions.

**Specific Comments:**

1. L 49: 'Pal et al' is repeated. There are several instances throughout the manuscript where the citations embedded in the sentence has repeated words.

**Answer:** We have modified all the similar situations in manuscript, as follows:

1) L 49 'The differential zero-crossing method proposed by Pal et al. (Pal et al.,1992)' is changed to ' The differential zero-crossing method proposed by Pal et al. (1992)'

2) L58 'Morille et al. ( Morille et al., 2007) determined the local maxima on both sides of the cloud peak as the cloud…' is changed to ' Morille et al. (2007) determined the local maxima on both sides of the cloud peak as the cloud'

3) L 60 'underestimated, respectively. Mao Feiyue (Mao et al., 2011)' is changed to ' underestimated, respectively. Mao Feiyue (2011)

4) L65 'Kollias et al. (Kollias et al., 2007) judged the SNR value' is changed to 'Kollias et al. (2007) judged the SNR value'

5) L67 'Clothiaux et al. ( Clothiaux et al.,1999) used 35 GHz millimeter wave cloud measuring radar' is changed to 'Clothiaux et al. (1999) used 35 GHz millimeter wave cloud measuring radar'

6) L170 'Referring to the empirical formula proposed by Riddle (Riddle et al., 1989)' is changed to 'Referring to the empirical formula proposed by Riddle (1989) '

7) L379 'Zhao et al. (Zhao et al., 2014) at the SGP site and Hailing Xie (Xie et al., 20217)' is changed to 'Zhao et al. (2014) at the SGP site and Hailing Xie (20217) '

2. L 115: Please use appropriate standard literature reference for the elastic backscattering lidar equation. For example: Measures, R.M., Laser remote sensing: Fundamentals and applications, Willey Publishers, 510 pp, 1984.

**Answer:** The reference 'Laser remote sensing: Fundamentals and applications, Willey Publishers, 510 pp, 1984' does not point out the standard lidar equation, so we refer to the standard radar equation in the reference 'Wandinger U.: Introduction to Lidar, Brooks/Cole Pub Co, doi:10.1007/0-387-25101-4_1, 2005.' as follows

$$P(\lambda,r) = P_0 \frac{c\tau}{2} A\eta \frac{O(r)}{r^2} \beta(\lambda,r) \cdot \exp\left[-2\int_0^r \sigma(\lambda,r)\,dr\right], \tag{1}$$

where λ is the wavelength of the emitted light, $r$ represents the detection distance, and $\beta(\lambda,r)$ and $\sigma(\lambda,r)$ are the atmospheric backscattering and extinction coefficients, respectively. $O(r)$ is the laser-beam receiver field-of-view overlap function,  $c$ is the speed of light, $P_0$ is the average power of a single laser pulse, $\tau$ is the temporal pulse length, $\eta$ is the overall system efficiency, and $A$ is the area of the primary receiver optics responsible for the collection of backscattered light.

3. L 133: Details on the wavelet function used should be mentioned here.

**Answer:** See the answer 4 below.

4. L 135: Complete description of the flow chart processes – variables are missing. For example, the

variables/symbols Rs, id, Pe, Ma, and Mi shown in figure 2 are nowhere defined.

**Answer:** After the statistical analysis of the system noise, we set $k = 4$ in this study. The algorithm flow chart of detecting cloud boundary by lidar is shown in Fig. 2. Usually, the moving average of $P_{new}(\lambda,r)$ of lidar echo signal is calculated to reduce the influence of random noise. However, the selection of a sliding window directly affects the signal quality. Therefore, $P_{new}(\lambda,r)$ is denoised by wavelet transform, threshold function is a soft threshold, wavelet base is sym7, and the number of decomposition layers is 5. Using wavelet function to reduce noise can avoid too much smoothing remove sharp signal changes due to clouds, and can also avoid the improper selection of moving average window. Obtaining cloud boundaries mainly includes three parts. The first part is signal preprocessing. $P_{new\_s}(\lambda,r)$ after wavelet de-noising is discretized based on the estimates of noise, and get useful signals $P_{new\_s1}(\lambda,r)$ and $P_{new\_s2}(\lambda,r)$. The second part is to enhance the signal to make the cloud signal sharper from the background noise and aerosol signal. Average signals $P_{new\_s1}(\lambda,r)$ and $P_{new\_s2}(\lambda,r)$ to obtain $P_{new\_sf}(\lambda,r)$. Ascending arrangement are conducted for $P_{new\_sf}(\lambda,r)$ and the new sequence $R_S$ and the corresponding index $id$ are recorded. The maximum and minimum $R_S$ are denoted as $Ma$ and $Mi$, respectively. By building a new mapping proportion coefficient $Pe(i)$, the enhanced signal $P_{new\_sp}(\lambda,r)$ is obtained. Obtain slope of baseline 1, and obtain baseline 2 based on this slope. Signals exceeding baseline 2 are regarded as candidate cloud signals as shown in Fig. 3b) and Fig. 4b). The third part is to extract cloud signal and realize boundary detection by combining the $SNR$ of echo signal. By fitting the echo signal slope in the height range of 15–20 km, the slope is used as the bottom slope to distinguish the cloud and aerosol layers (as shown by the magenta line in Fig. 3b and Fig. 4b). Without considering the bottom echo signal (0–2 km), the amplitude of the echo signal received by the lidar decreased with increasing detection height according to the fitted slope, as shown by the blue line baseline in Figs. 3b) and 4b). When the beam senses the presence of clouds, the amplitude of the echo signal will exceed the blue baseline.

5. L 138: What about the cases when high level clouds exist? It is well known that cirrus types of clouds occur high in the troposphere extending to the tropopause (on average ~17 km during summer over the subtropics).

**Answer:** In line 138, the 15-20km height range used to fit the slope is applicable to Xi'an region (in the data of the past two years, there are few clouds higher than 12km). When there are high-level clouds, the range of slope fitting should be the echo signal above the high-level clouds.

6. L 206: Do you mean the 2 min time-resolution lidar profiles are duplicated 24 times to make it look like 5 sec temporal resolution data? Please mention this clearly.

**Answer:** No, we perform linear interpolation on the lidar data within 2min, and keep its time-resolution consistent with MMCR, that is, the 2min one group of data becomes 2min 24 groups.

7. L 355: I don't think these are statistical rules. Please replace the term 'statistical rules' with 'logic-based' rules or something like that throughout the manuscript

**Answer:** L 355 'The statistical rules shown in Table 3' has been replaced by 'The cloud bottom height recording guideline in the Table3'. Other 'statistical rules' in the manuscript have been modified or replaced, and the specific details are given in the answer 15.

8. L 359: The case1 in Table 3, please be specific if you mean 'optically thin' or 'geometrical thin' cloud? If it is optically thin than both cloud top and base can unambiguously be determined from lidar.

**Answer:** L 359: 'Thin cloud' is changed to 'geometrical thin' cloud in Table 3.

9. L 376: Clouds above 8 km has highest frequency in autumn, and these are likely stratus and cumulus clouds? This sentence doesn't make any sense. Please refer to the WMO cloud classification – stratus and cumulus are low-level clouds those formed within 2 km above the surface level.

**Answer:** According to the expert opinion, we have re-defined the division of seasons (answer 12), so the observation data has been slightly adjusted, and the corresponding expression has changed. See the answer 10 below for details.

10. L 378: "height range of clouds is narrow, and the numerical range is wide"? Please re-write this sentence for more clarity mentioning the height range you are referring to.

**Answer:** For vertical distribution of cloud base, the first narrow peaks is boundary layer clouds ($\leqslant$ 1.5 km) , and the second peak is 2.5 ~ 3.5 km, and the third peak has a big range in vertical height, which is around 4.7-10 km a in spring. Fig.18 b) expresses that the cloud bottom height in summer is mainly distributed at 3-9.5km, indicating that the middle and high cloud may be dominated. The distribution of cloud bottom shows the bimodal, the first peak is the boundary layer cloud peak, and the second peak is located at 2.7-3.7 km and 3.6-8.3 km in autumn and winter, respectively.

11. L 383: Why the data presented in the figures showing vertical distribution of frequency of cloud occurrences are limited to 12 km? Or is this an underestimation of cloud top boundaries owing to the sensitivity of 8.6 mm radar? It is not uncommon to have high level clouds extending up to 15 km or more in the region of interest. Please present the results upto the tropopause level.

**Answer:** In July, 2021, the detection distance base of MMCR increased from 420 to 600, that is, the maximum detection range increased from 12.6km to 18km. We checked the echo data of MMCR (the maximum detection range is 18km) for 199 days one by one. Among them, only one day's data show that cirrus clouds existed at about 13km, and only four days' data show that the cloud top was slightly more than 12km. Therefore, our analysis of cloud boundary is limited to 12km. At the same time, we have added the specific time when the maximum detection range of MMCR changes in the manuscript. Add description as 'During the 12-month observation, the maximum detection range of MMCR has changed. From December 2020 to June 2021, the maximum detection range of MMCR is 12.6km, and then the maximum detection height is changed to 18km.'

12. L 390: Before discussing the result, it would be beneficial to briefly describe how normalized cloud cover is computed here. Also, indicate the months of the season – spring (MAM), and summer (JJA). How is the maximum cloud cover 2.46 in summer?

**Answer:**L 356-358 'The experimental data of 302 days (65 days in spring (January-March), 84 days in summer (April-June), 65 days in autumn (July-September) and 88 days in winter (October-December) observed in 2021 are classified and sorted out to ease the statistics and analysis of the variation characteristics of cloud boundary height', which is no solar term to define the season, so we re-describe it as 'From the above three cloud observation cases, it can be seen that MMCR has more advantages than lidar in detecting cloud-top boundaries. Therefore, when calculating the cloud boundary height distribution characteristics over Xi'an, we only counted the cloud top boundary height detected by the MMCR and considered it as the actual cloud top boundary. From December 2020 to November 2021, MMCR and lidar stored 302 d (7248 h) and 126 d (872.5 h) of observational data, respectively. During the 12-month observation period, the maximum detection altitude of the MMCR changed. From December 2020 to June 2021, the maximum detection range of MMCR is 12.6 km, and the maximum detection height is changed to 18 km. The total observation hours of MMCR and lidar for each month are shown in Fig. 15. The hours of lidar, MMCR, and simultaneous measurements are 872.5 h. In this study, the four seasons were defined as follows: spring from March to May (MAM), summer from June to August (JJA), autumn from September to November (SON), and winter from December to February (DJF).'

[Figure]

Fig. 19 Monthly variation in cloud frequency distribution and cloud cover from December 2020 to November 2021 a) monthly variation in the frequency of the number of cloud layers. b) monthly variation in cloud cover

MMCR defines cloud cover as the percentage of cloud obscuring sky field of vision. Cloud cover observation includes total, low, medium and high cloud cover. Total cloud cover refers to the total number of cloud cover in the sky during observation (Fig.18b shows the total cloud cover in every month). Generally, the sky is divided into 10 parts. When there is no cloud in the clear sky or less than 0.5 parts are covered, the cloud cover is zero. The cloud covers half of the sky and the cloud cover is 5. Cover the whole sky with clouds and the cloud cover is 10. Calculation steps: 1): divide

the cloud layer into high, medium and low families through the radial cloud base height. 2): average each cluster for 30 minutes. 3): Weighted Processing of data in 10 minutes to obtain the integrated cloud cover. Because the calculated cloud cover is a relative value, it does not mean the real cloud cover. Figure 18b shows that the cloud cover is the largest in April. Therefore, the cloud cover in April is set to 1, and the cloud cover in other months is calculated to represent the relative change trend of cloud cover in each month.

'The maximum cloud cover 2.46 in summer' is changed to 'It can be seen from the distribution of cloud cover in every month that there are relatively more cloud cover in summer and the least in winter, indicating that warm atmospheric conditions are more conducive to the formation and development of clouds. '

13. L 394: I suggest adding fig 18c showing the total monthly hours of lidar, radar and simultaneous measurements in this figure. This is essential to understand the reported cloud characterization.

**Answer:** The total observation hours of MMCR and lidar in each month are shown in Fig. 17. The hours of lidar, MMCR and simultaneous measurements is 872.5 hours. Considering the logic of the manuscript, we decided to put the Figure 17 in L364 in subsection 4.1.

[Figure]

Fig. 17 Monthly observation hours of lidar and MMCR

14. L 396: 'frequency change characteristics…'? This does not make any sense. As the figure caption says it is the frequency distribution of cloud boundaries observed over Xian in 2021.

**Answer:** L 396: 'Fig. 19 shows the frequency change characteristics of the cloud boundary vertical height distribution in 2021' is changed to 'As the Fig.20 caption says it is the frequency distribution of cloud boundaries observed over Xian from December 2020 to November 2021'.

15. L 424: Remove the word 'statistical'.

**Answer:** L 424: The word 'statistical' has been removed. The modified expression is 'Based on the analysis of the changes and distribution of cloud boundaries in Xi'an from December 2020 to November 2021.' At the same time, we have modified and replaced the word 'statistical' in other parts to make it closer to the aim of the manuscript. Such as 'Table 3 Statistical rules of cloud bottom boundary information' is changed to 'Table 3 Cloud bottom height recording guideline.' The word 'statistical' in L14 has also been removed.

**Response to reviewer #2 (in open discussion)**

**Major comments:**

This manuscript combines lidar and Ka-band millimeter-wave cloud radar (MMCR) to study the cloud macrophysical properties in Xi'an. The authors propose a local method for lidar and MMCR, but without enough details. It would be more interesting if detailed descriptions are added in this manuscript. The statistical analysis is kind of superficial and the English writing needs a full editing. It is difficult to follow for several times. Thus, I recommend a major revision and suggest the authors to rearrange this manuscript carefully.

We appreciate the reviewer's thoughtful review and constructive comments, which have greatly helped to enrich the details and improve the quality of the paper. We have added more details to the manuscript to make the logic of the article clearer and the details more perfect. The manuscript has been polished and modified by professional organizations, and I believe that English has been greatly improved. These comments have been revised and supplemented in the manuscript, and the responses to each comment are given below.

1. The title "Lidar and MMCR applied for the study on cloud boundary detection" indicates the manuscript will mainly focus on instruments and method, while the "statistical analysis of cloud distribution in Xi'an region" imply a systematically study for the local cloud distribution. This causes the keynote of the whole text not clear. Which part the authors want to focus, the method or statistical analysis? This would affect the structure of manuscript. Additional, both the method and statistical analysis of the manuscript as current form are not very clear.

**Response:** According to your requirements and suggestions, a series of modifications have been made in the manuscript. Reorganize the structure of the manuscript, reorganize the highlights and reorganize the language. In order to more clearly express the aim of the manuscript and consider the observation duration of MMCR and lidar, we have changed the title of the manuscript to "Detection and analysis of cloud boundary in Xi'an, China employing 35 GHz cloud radar aided by1064nm lidar".

2. The two flow charts of lidar and radar, i.e., Figure 2 and 6, are complex, but the text is too short. I can't tell if they are novel compared with previous methods. If the authors emphasize their method is well-improved, they should carefully introduce this part and show the difference and improvement from others'.

**Response:** We have modified figures 2 and 6 and added corresponding detailed descriptions as follows.

[revised manuscript text omitted]

3. The authors claimed several times "This study will combine the advantages of lidar and MMCR in detecting clouds". While it seems that the results are just simply calculated from the two instruments, respectively. I was hoping some more in-depth combination, like DARDAR for the space-born radar and lidar (Delanoe and Hogan, 2008), whose method is associated with the specific radar/lidar raw observational value.

**Response:** Dear reviewer, thank you for these valuable references, which we have carefully read and revised the manuscript accordingly based on your suggestions. We have cited these references at the corresponding places in the manuscript. As well as, these references have provided great help for my follow-up research. Obtaining accurate cloud information from echo signals is the premise of in-depth study of cloud micro parameters and analysis of special meteorological variation characteristics. In this study, we propose new methods for cloud boundary detection by lidar and MMC, and combined with special cases to verify and apply those methods. According to this main line of study, the research contents are full.

1). In the manuscript, based on the signal characteristic lidar and MMCR. We propose a new algorithm which suitable for accurately extracting cloud information from lidar echo signals. In order to improve the detection accuracy of MMCR, the cloud signals in Doppler spectra are identified in detail. The appropriate data quality control thresholds are established to effectively eliminate the floating debris echo signal.

2). Using lidar to identify cloud boundaries (cloud bottom and cloud top) is easily affected by aerosol and background noise. We extract the cloud signal effectively by wavelet change noise reduction, signal enhancement combined with the *SNR* of lidar echo signals. This method is not easy to be affected by noise and interfering signals, and also avoids the problem that cloud base and cloud top being overestimated or underestimated due to improper threshold selection. Compared with the previous research literature that directly uses the reflectivity of MMCR for cloud boundary recognition, the manuscript analyzes and calculates the noise level, $SNR_{min}$, and continuous common points from the initial Doppler spectra data of MMCR. Those make the recognition of meteorological signals more accurate.

3). Based on three special cases, we verified the proposed algorithm, and also clarified the detection advantages of lidar and MMCR under different conditions. Based on three special detection cases, the correctness and reliability of the proposed algorithm are verified, and the detection advantages of lidar and MMCR under different conditions are illustrated.

4). By processing and analyzing the accumulated observation data, a preliminary analysis of the changing characteristics of the cloud boundary is carried out in Xi'an.

So, it is unlikely that more research content needs to be added to the manuscript at present.

4. One-year observation might be too short for statistics analysis of cloud in section 4.2, especially only 302 days of MMCR and 126 days of lidar.

**Response:** At present, the amount of lidar and MMCR data in the manuscript is not enough to comprehensively and deeply analyze the cloud change distribution characteristics in Xi'an. Therefore, we have replaced or deleted 'statistics' in the text, and re-determined that the purpose of the manuscript is cloud boundary detection method research. The data analysis in the section 4.2 is the application of cloud boundary detection method. It provides a preliminary analysis for the distribution characteristics of cloud boundary in Xi'an. We also point out the contents to be studied in the future.

5. Most of the conclusions (line 413-423) are not new. There are many studies using collocated radar and lidar observation for cloud research, e.g. (Borg et al., 2011) (Dong et al., 2010) (Protat et al., 2011) and so on, which have shown similar results.

**Response:** L413-423 expresses some well-known advantages and disadvantages of lidar and MMCR for investigation of cloud, which makes the conclusions not detailed and in-depth. Therefore, we re-describe the conclusions (L 404 - 423), and also the points that can be improved in the follow-up of the manuscript are list.

Based on the observation data of lidar, a new algorithm is proposed which can effectively extract cloud signals. Compared with the previous method of identifying cloud bottom and cloud top from echo signals, the new method mainly obtains effective cloud signals through suppressing noise

signals and enhancing effective signals to realize cloud boundaries. The algorithm has two main characteristics: 1) in the signal preprocessing, wavelet transform is used for the original signal to avoid the defect of effective information loss caused by improper selection of smooth window; 2) The *SNR* of the signal is considered.

The cloud signals in Doppler spectra are effectively extracted by analyzing the noise level, $SNR_{min}$, and the continuous spectral points of Doppler spectra. The data quality control conditions for MMCR (reflectivity factor < -20 dBZ, spectra width >0.3 m/s and radial velocity < 0.2 m/s) were established by analyzing the characteristic of the interference of floating debris signals. By analysing the correlation of cloud bottom height between MMCR and lidar, and the cloud bottom height detection by MMCR with data quality control have a good agreement with lidar (the correlation coefficient is 0.803). Therefore, quality control is an important factor to improve signal accuracy of MMCR.

In this study, combined with the respective advantages of MMCR and lidar in cloud detection, the cloud cover and distribution of cloud boundaries characteristics are analyzed based on the observation data in Xi'an from December 2020 to November 2021.The result reveals that more than 34% of the clouds appear in the form of a single layer every month. The cloud cover was lowest in spring and highest in summer. The seasonal variation in cloud boundary height showed that the distribution characteristics of cloud boundaries in spring and summer were similar, and the frequency of high-level clouds in the range of 8–10 km was greater than autumn and winter. The stratiform clouds appearing below 3.5 km in autumn have the highest frequency, and high-level ice clouds or cirrus clouds above 8 km in winter are less likely to appear. The findings can provide a preliminary analysis of cloud boundary changes in Xi'an. If there are huge amounts of simultaneous observation data of lidar and MMCR, the comprehensive statistics and analysis of cloud macro and micro parameters can be realized, which can provide better support for the study of climate change characteristics in Xi'an.

**Minor Comments:**

**1**. Line 9, "he" should be "the"

**Answer:** Write error has been changed to "the".

**2**. Line 11, The SNR and SNRmin in the abstract should be explained and given the full description.

**Answer:** The *SNR* (Signal-to-noise ratio) is the ratio of lidar echo signal to noise signal, dimensionless. The $SNR_{min}$ refers to the noise ratio of the smallest measurable cloud signal in Doppler spectra signal. These have been described in the abstract.

**3**. Line 14, what does the "rules" mean?

**Answer:** The "rules" originally expressed the records of observation data in Table 3. We have changed Table 3 to "Cloud bottom height recording guideline". In L13-15 "Based on the advantages and disadvantages of the two devices in detecting cloud boundaries under different conditions, cloud boundary statistical rules are established to analyze the characteristics of cloud boundary changes in Xi'an in 2021'' is changed to "Based on the respective advantages of the two devices, the change characteristics of cloud boundary in Xi'an from December 2020 to November 2021 are analyzed with MMCR detection data as the main data and lidar data as assistant data.''

4. Line 33, what is "high change rate"

**Answer:** "However, the vertical structure distribution of clouds has great temporal and spatial heterogeneity and a high change rate, which leads to great challenges…." is changed to "However, the vertical structure distribution of clouds has great temporal and spatial heterogeneity, which leads to great challenges…."

5. Line 35-36, remove "direction", …has always been important for cloud physics.

**Answer:** "Notwithstanding, research on the characteristics of cloud vertical structures has always been an important direction of cloud physics research. " is changed to "Notwithstanding, research on the characteristics of cloud vertical structures has always been an important for cloud physics."

6. Line 50," dP/dr", what is P and r, what is "negative to positive", you mean the value of dP/dr, from negative to positive? Please rephrase this sentence.

**Answer:** Re-describe L50 as, "Calculation of *dP/dr* using lidar backscattering intensity *P* and range *r*, and the first derivative of backscatter intensity *dP/dr* changes sign from negative to positive and this zero crossing is cloud bottom. "

7. Line 54, what is "detail debugging"

**Answer:** The 'detail debugging' means that the threshold method needs to be changed according to experience in the calculation process. Unclear expression has been modified in the manuscript.

Line 54, "It is easily affected by noise, and some indicators must be introduced in the specific implementation process to determine the cloud boundary through complex detail debugging, which brings certain difficulties to accurate cloud boundary detection" is changed to "It is easily affected by noise, and restrictive parameters must be introduced in the specific implementation process to determine the cloud boundary by adjusting the parameters, which brings certain difficulties to accurate cloud boundary detection"

8. Line 59, "but the cloud bottom and cloud top detected by this method will be overestimated and underestimated respectively". Does this mean the method would miss some part of cloud, i.e., detect some real cloud signal as noise? This manuscript really needs complete English editing.

**Answer:** "but the cloud bottom and cloud top detected by this method will be overestimated and underestimated respectively." means that the real signal at the cloud bottom may be considered as noise, and the real signal at the cloud top may be miss. It has been re-expressed as ", but this method takes some real signals at the cloud bottom as noise and miss information at the cloud top, and resulting in overestimation and underestimated of cloud base and cloud top height respectively."
The English language of the manuscript has been polished and modified by professional institutions.

9. Line 65, what is "library". Line 65-67 is different to understand.

**Answer:** The "distance library" in the line 65 is changed to "gate".
Line 65-67 is re described as: "Kollias et al. (2007) judge step by step from the bottom to the top of the reflectivity. If the *SNR* of 9 consecutive distance gates is greater than the set threshold, these gates represented as cloud signals. Otherwise, they are non-cloud signal."

10. Figure 1. The area could be lager, at least shows some of the "Guanzhong Basin", "Weihe River Basin", "Loess Plateau", "Qinling Mountains" as you described in line 88-90. What does the while line mean? Is it really necessary to show the negative elevation in your color bar?

**Answer:** The white line in the original diagram originally denotes the Xi'an Region. The revised Fig. 1 contains the "Guanzhong plain", "Weihe River", "Loess Plateau" and "Qinling Mountains" as follow. In the Fig. 1, the black line represents Shaanxi Province, the dark blue represents the Yellow River, and the wathet blue represents the Weihe River.

[Figure]

Fig. 1. Geographical coverage of Shaanxi Province (105 °29'-111 °15'E, 31 °42'-39 °35'N). The red dot indicates the location of the Jinghe National Meteorological Station in Xi'an.

11. Line 99, what is "HT101"?

**Answer:** TH101 is the model of the MMCR

12. Line 113-114 is difficult to understand.

**Answer:** "When using lidar for detection, the laser beam propagates in a clear atmosphere, and the received echo power continuously decreases with increasing detection height. However, the beam into the clouds (or aerosols, etc.), the echo power increases suddenly and becomes stronger at a distance above the cloud bottom. The lidar equation owing to elastic backscattering can be written as (Motty et al., 2018)," was re-described as "The lidar equation owing to elastic backscattering (Motty et al., 2018) can be written as,"

13. Line 116 and line 120, should the $N_{bcak}$ be $N_{back}$?

**Answer:** "$N_{bcak}$" is changed to "$N_{back}$"

14. Figure 2. "yes" and "no" may be marked in the wrong place. They should be marked after a judgment statement, i.e., ">", "<" or "==", rather than equations. The symbols in the text should be explained. What is "sort", "Pe"? What is the relationship between the three main boxes? It is hard to follow just from line 134-135.

**Answer:** Sorry, "yes" and "no" are misplaced in the flow chart 2. The revised figure 2 and text description are shown in the Major comments two.

15. Figure 3. The box, axis, tick should be black. The other figures in the manuscript should be changed too. Why the time in title is different with the time in figure? What is the unit of x axis? I notice there are some signal below the blue base line in figure b, especially below cloud base height, around 8 km, 6 km and 4 km. The slope is obviously different with the fitting slope. Does this influence your detection? What is the vertical dashed line in figure c?

**Answer:** The box, axis, tick of all figures have changed black in the manuscript. The time of the legend in the figure is correct, and "Fig. 3 Detection results of the lidar at 19:15 on March 4, 2021" is wrongly written due to negligence, and has been modified in the paper. The situation in Fig.3b) does not affect the identification of subsequent cloud boundaries, and the signals below the blue baseline (especially at 8 km, 6 km and 4km) are considered as aerosol signals or interference information and will be eliminated. Fig.3c) *S/N* in Shannon formula is the power ratio of signal to noise, which is a dimensionless unit. The blue vertical dotted line is only a schematic auxiliary line in Fig.3c), indicating that the *SNR* of the cloud should be greater than 5 in this case.

[Figure]

Fig. 3 Detection results of the lidar at 12:13 on March 5, 2021: a) $P_{new\_sf}$ of the 1064 nm signal, b) $P_{new\_sp}$ of the 1064 nm signal, c) SNR of $P_{new\_sf,}$ and d) cloud information detected

16. Figure 6. The "thresh of XXX" should be "Larger/Smaller than thresh of XX". Generally, it should be a judgment statement.

**Answer:** Figure 6 has been modified. The revised figure 6 and text description are shown in the Major comments two.

17. Line 176, what is $N_{ts}$?

**Answer:** $N_{ts}$ represents the threshold value of continuous spectral points. The "$N_{ts}$" has been described in the manuscript.

18. Line 184-185, please do not use both ">" and "less than" in one sentence. What is the unit of "velocity" and "velocity spectrum width"? And why you choose such thresholds?

**Answer:** "As shown in Fig. 6b), when the subjective echo intensity Z<-20 dBZ, the absolute value of radial velocity is less than 0.2, and the velocity spectrum width >0.3 is used as the threshold for removing nonmeteorological information, the expected data quality control requirements can be met." is changed to "As shown in Fig. 6b), when reflectivity Z<-20 dBZ, the absolute value of radial velocity < 0.2 m/s, and the velocity spectra width >0.3 m/s are used as the threshold for removing non-cloud information, the expected data quality control requirements can be met.''

19. Figure 7. Is the unit of velocity spectrum width in figure c "m/s"? Figure a, echo emissivity factor is the same as "reflectivity factor"?

**Answer:** Yes, the unit of velocity spectra width is m/s in Figs7. c). "Figure7 a), echo emissivity factor" and "reflectivity factor" in Figs 7. a) and d) are consistent, and they are uniformly expressed as 'reflectivity' in the manuscript.

[Figure]

Fig. 7 Meteorological signals of MMCR at 22:44 on June 8, 2021. a) reflectivity, b) radial velocity, c) velocity spectra width, d) echo emissivity factor after quality control

20. Line 212, What is "time-height-indicator information"? Do you mean "vertical profile"?

**Answer:** No, "time height indicator information" is used to describe the long-term observation results in Fig. 8." The sentence is re-described in the manuscript.

"According to the data method described in Section 3.1, the SNR of $P_{new\_sf}$ and $P_{new\_sp}$ of the echo signal of the lidar @1064 nm are obtained time-height-indicator information (THI) and are shown in Figs. 8a) and 8b)." is changed to "According to the data method described in Section 3.1, we can obtain cloud change information of time-height-indicator (THI) for *SNR* of $P_{new\_sf}$ and $P_{new\_sp}$ of lidar @1064nm with a duration of 7 hours, as shown in Figs. 8a) and 8b)."

21. Line 214-215, "After 05:00, the cloud layer developed deeper". Does this infer from Figure 9, the MMCR observation? It would be clearer if you combine Figure 8, 9 and 10 together to see the difference of the two instruments. Same as Fig 11-13, and Fig 14, 15.

**Answer:** Yes, this phenomenon can be seen from Figure 9 that the clouds are developing deeply. We have combined figure 8 and Figure 9 in the original text and described them again. It can be seen from Fig.8 d) that the cloud layer developed deeper after 5:00, and the laser beam penetrated 0.1 km into the cloud layer and was quickly attenuated.

[Figure]

Fig. 8 The THI of echo signal of the lidar and MMCR on March 4 to 5, 2021. a) *SNR* of the 1064 nm signal, b) $P_{new\_sp}$ of the 1064 nm signal, c) cloud information detection results of the lidar, d) reflectivity of the MMCR without quality control, e) reflectivity the MMCR with quality control (dotted line indicates rainfall time)

22. Line 216, "Rainfall begins at 06:00", how do you get the time of rainfall, do you have rain gauge or other observations? Please explain this in Section 2.

**Answer:** We checked the time of rainfall recorded by microwave radiometer, which is close to MMCR. The record of rainfall time has been described in the manuscript.

23. Figure 8. What is the stripe in figure b around 23:00-01:00? Does this affect your detection results? What does the "SNR>5.2" in figure c stand for?

**Answer:** The stripes around 23:00-01:00 in Fig. 8b are caused by the instability of the laser seed, which causes slight fluctuations in the emitted light energy, but this does not affect lidar detection of clouds, nor does it affect the recognition of cloud boundary. "SNR >5.2" in Fig.8 c) indicates that we get the cloud boundary shown in Fig. 8c), we only retain the effective data lattice with SNR >5.2 (regardless of the underlying signal saturation region) in Fig. 8a).

25. Line 232, "the cloud layer starts at 03:00", does this mean the signal before 03:00 is not cloud?

**Answer:** No. The information displayed is cloud signal from 19:00 to 06:00 in Fig.9b). "From the THI of the echo reflectivity of the cloud, the cloud layer starts at 03:00 and gradually develops from 7 km to 12 km (the lidar signal fails to show this detail)." is changed to "According to the echo emissivity factor of the MMCR, from 03:00 to the end of observation, the cloud layer developed deeper, the cloud bottom height gradually decreased from 7 km to 300m, and the cloud top height developed to ~12 km (the lidar signal fails to show this detail). "

26. Line 253-254, "From the characteristic distribution of the $P_{new\_sp}$ signal in Fig. 11b), the low-level cloud rained from 18:30 to 18:45", how does this be concluded, just by the sudden decree of cloud base?

**Answer:** In the observation experiment at 18:30 on March 4, 2021, we felt that there were small showers on the ground and the duration was ~10 mins. Then we checked the rainfall time recorded by microwave radiometer (recording every 2 min), and the specific rainfall period was 18:30~18:45 CST and the precipitation reached the ground. At the same time, the radial velocity of MMCR showed that the velocity reaches ~-4m/s in this period.

27. Line 273, "During the period from 15:00 to 01:00", where is "15:00" in figure 12?

**Answer:** Sorry, I mistakenly wrote 17:00 as 15:00 due to negligence. "During the period from 15:00 to 01:00…" is changed to "During the period from 17:00 to 01:00…"

28. Figure 13, Could you please at least use one specific color/line style/marker to represent one property (cloud base or top/first or second layer/lidar or MMCR)?

**Answer:** The changed Figure 13 as below.

[Figure]

Fig. 13 Cloud boundary detected by the lidar and MMCR from March 4 to 5, 2021

29. Line 294, "Case three studies of precipitating cloud", the figures of case one and two are also have been marked with rainfall. If you want to discuss precipitating cloud separately, the case one and two should be non-precipitating cloud.

**Answer:** We changed the objectives of the three study cases to the following,

"1) Case one studies of double-layer clouds" is changed to "1) First case study period".

"2) Case two studies of double-layer clouds" is changed to "2) Second case study period".

"3) Case three studies of precipitating cloud" is changed to "3) Third case study period".

30. Line 310, what is "rain storage"?

**Answer:** The "rain storage" means "rain virga". "As the observation time progresses, the phenomenon of rain storage (reflectivity >-15 dBZ) occurs in the cloud" is changed to "As the observation time progresses, the phenomenon of rain virga (reflectivity >-15 dBZ) occurs in the cloud"

31. Figure 15. How the cloud base height being determined for precipitating cloud, such as after 11:00? I don't think the cloud base height around 0 km is appropriate. This may explain why the cloud base height in figure 19 has a such huge peak at lower level.

**Answer:** When rainfall is slightly more intense, neither laser radar nor millimeter radar achieves an accurate assessment of cloud base height (visiting a balloon perhaps achieves an approximate detection but is not part of this paper's research content). Figure11: the cloud base height after 11:00 is 0.27 km instead of 0 m. In the Fig. 15, because there are a larger number of plotted points, the cloud bottom height around 0 km is appropriate.

[Figure]

Fig. 15 Cloud boundary detected by the lidar and MMCR on March 10, 2021

32. Line 338, the 126 days of lidar observations seems too short for one year. Can the authors explain why is that? Is there any issue of the lidar, if so, does this issue affect the observed results?

**Answer:** The main task of the lidar in the manuscript is to monitor special weather changes, so the data volume is only 126 days in 2021. This does not affect our cloud boundary analysis for the whole year, because MMCR data are mainly used in cloud boundary analysis.

33. Line 341-342, "we plan to employ MMCR data to replace the data of periods when the lidar is not running" What do you mean by "replace"? You mean the MMCR data are only useful when lidar is not running? Generally, I am not sure the purpose of Figure 16 and Table 3. "bottom of MMCR is blurred" in Table 3, what does this mean? Are the results of table 3 accomplished by manual selection?

**Answer:** According to the results discussed in the previous chapters, lidar has more advantages than MMCR in cloud bottom detection. Therefore, lidar (detecting cloud bottom) and MMCR (detecting cloud top) can be combined to detect cloud boundary (cloud bottom and cloud top), but considering the continuous observation time of lidar, it is not enough to analyze the change of cloud bottom all the year. Therefore, we analyzed the correlation between the cloud bottom detected by MMCR with quality control and lidar, and the correlation coefficient is 0.803. Therefore, the cloud bottom height during the period when the lidar is not running is provided by MMCR to realize the annual cloud boundary change in Xi'an.
Figure 16 mainly shows that the cloud bottom height is good agreement with the lidar and MMCR with data quality control. Therefore, when the lidar is not operational, the cloud bottom information can be provided by MMCR.

The "bottom of MMCR is blurred" in Table 3 indicates that the MMCR cannot accurately identify the cloud bottom in light rain or drizzle. "Bottom of MMCR is blurred" is changed to "bottom of MMCR is invalid".

The data selection in Table 3 is provided by our developed algorithm.

34. Line 379, "20217" should be "2017".

**Answer:** Sorry, "20217" has been changed to "2017".

35. Line 385-386, "The months with the largest (96%) and smallest (42%) cloud occurrence frequencies are August and December, respectively." Does this mean the Jinghe National Meteorological Station are nearly covered by cloud during the whole month of August? Does it make any sense?

**Answer:** "The months with the largest (96%) and smallest (42%) cloud occurrence frequencies are August and December, respectively." indicates that 96% and 42% of all profiles detected in the 22 days of August and 30 days of December contain cloud profiles, indicating that the frequency of cloud formation is the highest and lowest in August and December respectively. This number of '96%' is relatively large, and Line l389 explains why "96%" is large.

36. Line 390-391 and figure 18b, how the "normalized monthly distribution" be calculated? "the minimum cloud amount is 0.65 in spring and the maximum is 2.46 in summer", how do these two numbers be inferred?

**Answer:** MMCR defines cloud cover as the percentage of cloud obscuring sky field of vision. Cloud cover observation includes total, low, medium and high cloud cover. Total cloud cover refers to the total number of cloud cover in the sky during observation (Fig.18b shows the total cloud cover in every month). Generally, the sky is divided into 10 parts. When there is no cloud in the clear sky or less than 0.5 parts are covered, the cloud cover is zero. The cloud covers half of the sky and the cloud cover is 5. Cover the whole sky with clouds and the cloud cover is 10. Calculation steps: 1): divide the cloud layer into high, medium and low families through the radial cloud base height. 2): average each cluster for 30 minutes. 3): Weighted Processing of data in 10 minutes to obtain the integrated cloud cover. Because the calculated cloud cover is a relative value, it does not mean the real cloud cover. Figure 18b shows that the cloud cover is the largest in April. Therefore, the cloud cover in April is set to 1, and the cloud cover in other months is calculated to represent the relative change trend of cloud cover in each month.
'the minimum cloud amount is 0.65 in spring and the maximum is 2.46 in summer' is changed to 'It can be seen from the distribution of cloud cover in every month that there are relatively more cloud cover in summer and the least in winter, indicating that warm atmospheric conditions are more conducive to the formation and development of clouds.

We have carefully revised the manuscript base on the opinions of reviewers and public discussion. The modified part has been marked with blue font. The specific modification list is as follows:

**Title and authors**

| Number | original manuscript | Number of line | revised manuscript | Number of line |
|---|---|---|---|---|
| 1 | Lidar and MMCR applied for the study on cloud boundary detection and the statistical analysis of cloud distribution in Xi'an region | line 1-2 | Detection and analysis of cloud boundary in Xi'an, China employing 35GHz cloud radar aided by 1064nm lidar | line 1-2 |
| 2 | Yun Yuan , Huige Di *, Tao Yang , Yuanyuan Liu , Qimeng Li, Qing Yan, Dengxin Hua* | line 3-4 | Yun Yuan , Huige Di *, Yuanyuan Liu ,Tao Yang , Qimeng Li, Qing Yan, Wenhui Xin, Shichun Li,  Dengxin Hua* | line 3-4 |

**0) Modification the part of Abstract**

[revised manuscript text omitted]

| | | | | |
|---|---|---|---|---|
| | process to determine the cloud boundary through complex detail debugging, which brings certain difficulties to accurate cloud boundary detection. | | 1995). They are easily affected by noise, and some indicators must be introduced in the specific implementation process to determine the cloud boundary by changing the experience threshold frequently during calculation, which causes difficulties in accurate cloud boundary detection. | |
| 14 | , but the algorithm | line 56 | However, the algorithm | line 62 |
| 15 | WCT (wavelet covariance transform) | line 257 | wavelet covariance transform method, | line 63 |
| 16 | Morille et al. ( Morille et al., 2007) | line 58 | Morille et al. (2007) | line 63 |
| 17 | detected by this method will be overestimated and underestimated, respectively. | line 59-60 | but this method takes some real signals at the cloud bottom as noise and miss information at the cloud top, and resulting in overestimation and underestimated of cloud base and cloud top height respectively. Mao (2011) | line 64-66 |
| 18 | , and realized the | line 61 | , and detected the | line 67 |
| 19 | to detect the cloud boundary ( Haper et al., 1966; Hobbs et al.,1985; Platt et al., 1994; Brown et al., 1995). Kollias et al. (Kollias et al., 2007) judged the SNR value of a 5×5 grid centered on a distance library. If the SNR of more than 9 consecutive libraries reaches the threshold, the distance library is a cloud signal; otherwise, it is judged as a noncloud signal. | line 64-67 | used to detect the cloud boundary (Hobbs et al., 1985; Platt et al., 1994). Kollias et al. (2007) judge step by step from the bottom to the top of the reflectivity. If the *SNR* of more than nine consecutive distance gates reaches the set threshold, these gates represented as cloud signals; otherwise, it is deemed a noncloud signal. | line 70-72 |
| 20 | Due to the existence of certain ground object | line 69 | The existence of certain ground object | line 74 |
| 21 | in the lower atmosphere, it will interfere with the real cloud echo signal | line 70 | in the lower atmosphere interferes | line 75 |
| 22 | , it will result in large errors in the detection of cloud boundaries. | line 73 | large errors in the detection of cloud boundaries result. | line 78-79 |
| 23 | At present, | line 75 | Currently, | line 81 |
| 24 | Sasse et al., 2001) | line 76 | Sasse et al., 2001; Borg et al., 2011; Delanoe and Hogan, 2008) | line 82-83 |
| 25 | in this paper, | line 81 | in this study, | line 87 |

**2) Modification the part of Observation and Instrument**

| Number | original manuscript | Number of line | revised manuscript | Number of line |
|--------|---------------------|----------------|--------------------|----------------|
| 1 | Xi'an (107.40 ~ 109.49 E and 33.42 ~ 34.45 N) | line 88 | Xi'an City (107°.40'-109°.49'E, 33°.42-34°.45'N), Shaanxi Province (105°29'-111°15'E, 31°42'-39°35'N) | line 94 |
| 2 | of sky clouds. Fig. 1 shows the topography of Xi'an and the site location of the Jinghe Meteorological Station. | line 93-94 | of clouds. Black line represents Shaanxi Province, dark blue represents the Yellow River, wathet blue represents the Weihe River, and red dot indicates the location of the Jinghe National Meteorological Station in Fig. 1. | line 99-101 |
| 3 | Figure 1 | line 95-97 | Figure 1 | line 102-104 |

**3) Modification the part of Method**

| Number | original manuscript | Number of line | revised manuscript | Number of line |
|---|---|---|---|---|
| | in the actual observation process | 107 | during the actual observation | 107 |
| | of echo signals. | 110 | in the echo signals. | 117 |
| 1 | When using lidar for detection, the laser beam propagates in a clear atmosphere, and the received echo power continuously decreases with increasing detection height. However, the beam into the clouds (or aerosols, etc.), the echo power increases suddenly and becomes stronger at a distance above the cloud bottom. The lidar equation owing to elastic backscattering can be written as (Motty et al., 2018), | line 112-115 | The lidar equation owing to elastic backscattering (Wandinger, 2005; Motty et al., 2018) can be written as, | line 119 |
| 2 | Add formula | | $$P(\lambda,r)=P_0\frac{c\tau}{2}A\eta\frac{O(r)}{r^2}\beta(\lambda,r)\cdot\exp\left[-2\int_0^r\sigma(\lambda,r)dr\right] \quad (1)$$ | line 110 |
| 3 | where $\lambda$ is the wavelength of the emitted light, $r$ represents the detection distance, and $C$ is the system constant, which is determined by the laser energy, the receiving area of the telescope, the quantum efficiency of the detector, etc. $\Delta r$ is the detection range resolution of the system, and $\beta(\lambda,r)$ and $\sigma(\lambda,r')$ are the atmospheric backscattering coefficient and atmospheric extinction coefficient, respectively. $N_{bcak}(\lambda,r'')$ is the background noise received by the system. $E(\lambda,r)$ represents noise brought to the detection system obtained by calibration.

  To avoid amplifying the high-level noise signals, we do not perform the distance square correction Eq. (1) and directly process it as follows: | line 17-123 | where $\lambda$ is the wavelength of the emitted light, $r$ represents the detection distance, and $\beta(\lambda,r)$ and $\sigma(\lambda,r)$ are the atmospheric backscattering and extinction coefficients, respectively. $O(r)$ is the laser-beam receiver field-of-view overlap function, $c$ is the speed of light, $P_0$ is the average power of a single laser pulse, $\tau$ is the temporal pulse length, $\eta$ is the overall system efficiency, and $A$ is the area of the primary receiver optics responsible for the collection of backscattered light.
Considering the influence of the background noise and response noise of the photomultiplier detector, Eq. (1) can be further expressed as | line 121-127 |
| 4 | $$P_{new}(\lambda,r)=\frac{P(\lambda,r)-E(\lambda,r)-N_{bcak}(\lambda,r'')}{C\cdot\Delta r}$$ | line 124 | $$P(\lambda,r)=C\cdot\frac{\Delta r}{r^2}\cdot\beta(\lambda,r)\cdot\exp\left[-2\int_0^r\sigma(\lambda,r)dr\right]+E(\lambda,r)+N_{back}(\lambda,r')$$ | line 128 |

| 5 | For ground-based lidar, the echo signal at a certain height range (>15 km in this study) can be considered background and electrical noise, $N_{bcak}(\lambda,r'')$ can be estimated with the signal within this range, and the standard deviation of the noise within the distance range is calculated: | line 125-127 | where $C$ is the system constant, which is determined by the laser energy, receiving area of the telescope, and quantum efficiency of the detector. $\Delta r$ is the detection range resolution of the system. $N_{back}(\lambda,r')$ is the background noise received by the system. $E(\lambda,r)$ represents the noise introduced to the detection system by calibration.

To avoid amplifying the high-level noise signals, we do not perform distance square correction in Eq. (2) but directly process it as follows: | line 129-131 |
|---|---|---|---|---|
| 6 | $$P_{new}(\lambda,r)=\frac{P(\lambda,r)-E(\lambda,r)-N_{bcak}(\lambda,r'')}{C\cdot\Delta r}$$ | line 124 | $$P_{new}(\lambda,r)=\frac{P(\lambda,r)-E(\lambda,r)-N_{back}(\lambda,r')}{C\cdot\Delta r}$$ | line 134 |
| 7 | (>15 km in this study) can be considered background and electrical noise | line 125-126 | (>15 km in this study applied to the Xi'an region) can be considered as molecular scattering | line 135-136 |
| 8 | is calculated: | line 127 | is calculated as follows: | Line137 |
| 9 | where $x$ is | line 129 | where $x$ denote | line 139 |
| 10 | we set $k=4$ in this paper. Usually, the moving average of $P_{new}(\lambda,r)$ is performed to reduce the influence of random noise. However, the selection of the sliding window directly affects the quality of the signal. Therefore, in this paper, we use the soft-threshold wavelet denoising method to process $P_{new}(\lambda,r)$ to obtain $P_{new\_s}(\lambda,r)$. To avoid atmospheric turbulence and noise interference, $P_{new\_s}(\lambda,r)$ is processed in one step according to the algorithm flow in Fig. 2, and the enhanced signal $P_{new\_sp}(\lambda,r)$ is obtained, as shown in Fig. 3b) and Fig. 4b). The cloud signal is prominently increased from the background noise and the aerosol signal compared to Fig. 3a) and Fig. | line 131-138 | we set $k = 4$ in this study. The algorithm flow chart of detecting cloud boundary by lidar is shown in Fig. 2. Usually, the moving average of $P_{new}(\lambda,r)$ of lidar echo signal is calculated to reduce the influence of random noise. However, the selection of a sliding window directly affects the signal quality. Therefore, $P_{new}(\lambda,r)$ is denoised by wavelet transform, threshold function is a soft threshold, wavelet base is sym7, and the number of decomposition layers is 5. Using wavelet function to reduce noise can avoid too much smoothing remove sharp signal changes due to clouds, and can also avoid the improper selection of moving average window. Obtaining cloud boundaries mainly includes three parts. The first part is signal preprocessing. $P_{new\_s}(\lambda,r)$ after wavelet de-noising is discretized based on the estimates of noise, and get useful | line 141-155 |

[revised manuscript text omitted]

| | | | | |
|---|---|---|---|---|
| | | | proportional to the square of the particles. From 19:00 to 00:00 CST, cirrus cloud transition to altostratus, where size of cloud particles increases in the form of collision and finally produces precipitation. In this process, the lidar beam entering the cloud is attenuated, but MMCR has a good advantage in cloud-top detection. | |
| 9 | Figure 10 | line 245-246 | Figure 9 | line 270-271 |
| 10 | 2) Case two studies of double-layer clouds | line 247 | 2) Second case study period | line 272 |
| 11 | From March 4 to 5, 2021, | line 248 | From 4 to 5 March 2021, | line 273 |
| 12 | Fig. 11a) and Fig. 1b). These THIs display | line 250 | Figs. 10a) and 10b). These THIs reveal | line 275 |
| 13 | during the observation process. | line 251 | during the observation period. | line 276 |
| 14 | During the period from 17:00 to 01:00, there is a relatively weak $P_{new\_sp}$ signal | line 258 | From 17:00 to 01:00 CST, there was a relatively weak $P_{new\_sp}$ signal | line 283 |
| 15 | the echo reflectivity of MMCR | line 269 | the reflectivity factor of the MMCR | line 295 |
| 16 | From the joint observation results | line 271 | The joint observation results | line 297 |
| 17 | During the period from 15:00 to 01:00, | line 273 | From 17:00 to 01:00 CST | line 299 |
| 18 | is obviously better than | Line274 | was markedly better than | line 299 |
| 19 | Figure 11 and Figure 12 | line 266, 284 | Figure 10 | line 290-292 |
| 20 | Based on the cloud signals (Fig. 11c and Fig. 12b) jointly observed by the lidar and MMCR, the height distribution of the double-layer cloud boundaries is detected, as shown in Fig. 13. From the cloud boundary height distribution, it can be seen | line 286-288 | The height distribution of the double-layer cloud boundaries was detected based on the cloud signals (Fig. 10c and Fig. 10e) jointly observed by lidar and MMCR, as shown in Fig. 11. The cloud boundary height distribution shows | line 309-310 |
| 21 | lidar has total supremacy in detecting the information of thin clouds. | line 291 | lidar is superior in detecting thin cloud information. | line 214 |
| 22 | Figure 13 | line 292-293 | Figure 11 | line 315-316 |
| 23 | On March 10, 2021, | line 295 | On 10 March 2021 | line 318 |
| 24 | the echo reflections of | line 299 | reflectivity factor of | line 322 |
| 25 | which makes | line 301 | which simplifies | line 324 |
| 26 | Figure 14 | line 303-306 | Figure 12 | line 339-342 |
| 27 | rain storage (>15 dBZ) | line 310 | rain virga (> -15 dBZ) | line 329 |
| 28 | while the millimeter | line 311 | whereas the millimeter | line 330 |
| 29 | There was a drizzle falling from 09:00 to 10:45, | line 317 | A drizzle fell from 09:00 to 10:45 CST, | line 336 |

| 30 | Figure 15 | line 30-321 | Figure 13 | line 343-344 |
|----|-----------|-------------|-----------|---------------|
| 31 | the beam of the lidar will be seriously attenuate | line 329 | the more severely the beam of the | line 352 |
| 32 | the complete cloud information at this time. | line 331 | complete cloud informatio | line 354 |
| 33 | 4.2 Statistics and analysis of cloud boundary distribution characteristics in Xi'an | line 336 | 4.2 Analysis of cloud boundary distribution characteristics in Xi'an | line 359 |
| 34 | In 2021, the Lidar and MMCR radar conducted cloud observation experiments at the Jinghe meteorological station, in which the MMCR accumulated 302 days of data (7248 hours in total) and the lidar observed 126 days (872.5 hours in total). Due to some unavoidable external reasons, the lidar failed to carry out the observation experiment at the same time as the MMCR. To further analyze the changes in the height distribution of cloud boundaries in Xi'an in 2021, we plan to employ MMCR data to replace the data of periods when the lidar is not running. | line 337-342 | To further analyse the changes in the height distribution of cloud boundaries in Xi'an, we plan to use MMCR and lidar data for cloud boundary analysis. | line 360-361 |
| 35 | principles and detection algorithms | line 348 | principles and algorithms | line 367 |
| 36 | Figure 16 | line 350-351 | Figure 14 | line 369-370 |
| 37 | From the above three cloud observation cases, it can be seen that MMCR has more advantages than lidar in detecting cloud top boundaries. Therefore, when calculating the cloud boundary height distribution characteristics over Xi'an in 2021, we only count the cloud top boundary height detected by MMCR and take it as the actual cloud top boundary. The statistical rules shown in Table 3 are established for the statistics of cloud bottom boundary information. The experimental data of 302 days (65 days in spring (January-March), 84 days in summer (April-June), 65 days in autumn (July-September) and 88 days in winter (October-December) observed in 2021 are classified and sorted out to ease the statistics and analysis of the variation characteristics of cloud boundary height. | line 352-358 | From the above three cloud observation cases, it can be seen that MMCR has more advantages than lidar in detecting cloud-top boundaries. Therefore, when calculating the cloud boundary height distribution characteristics over Xi'an, we only counted the cloud top boundary height detected by the MMCR and considered it as the actual cloud top boundary. From December 2020 to November 2021, MMCR and lidar stored 302 d (7248 h) and 126 d (872.5 h) of observational data, respectively. During the 12-month observation period, the maximum detection altitude of the MMCR changed. From December 2020 to June 2021, the maximum detection range of MMCR is 12.6 km, and the maximum detection height is changed to 18 km. The total observation hours of MMCR and lidar for each month are shown in Fig. 15. The hours of lidar, MMCR, and simultaneous measurements are 872.5 h. In this study, the four seasons were defined as follows: spring from March to | line 371-380 |

[revised manuscript text omitted]

**Added 5 references, indicated in blue font**

---

## Referee Report (RR1)

Review for revised manuscript of "Detection and analysis of cloud boundary in Xi'an, China employing 35GHz cloud radar aided by 1064nm lidar" by Y. Yuan et al.

I believe the authors responded to the review comments in detail and the revised manuscript can be considered for a publication. There are some minor comments that may need to be further considered.

1. Line 64-65: "take" needs to be replaced with "mistake", "underestimated" should be "underestimation".

2. Line 105: Please check "HT101", which is "TH101" in the response.

3. Line 145: What is the full name of "sym" in "sym7"? It should be clarified.

4. Line 153: Please add detailed description of the "baseline 1"and the "baseline 2" in the manuscript, and what is the difference between them?

5. Figure 3: Fig. 3c shows the vertical profile of SNR. However, "S" is marked on the horizontal axis. I'm confused whether it is the abbreviation of SNR and why the unit of S is "N".

6. Line 184-185: "two parts" appears twice in this sentence.

7. Line 203-205: Why you choose reflectivity of 20dBZ, velocity of 0.2m/s and spectra width of 0.3m/s as the thresholds?

8. Line 214: There is no "Fig. 6b" in Figure 6, please check.

9. Line 251: "The microwave radiometer accurately records the rainfall time" is mentioned, but the "similar to the following" is difficult to understand. The specific rainfall time recorded by the microwave radiometer should be illustrated.

10. Line 258: The dotted line in Figure 8 is blue instead of "black". The color bars for a), b) and c) in Fig 8, Fig.10 and Fig.12 should be labeled with unit.

11. Figure 13: In the response version, the authors claim that "because there are a larger number of plotted points, the cloud bottom height around 0 km is appropriate", one should note that the bottom echo signal is not the height of the cloud base for precipitating cloud. If this height is treated as the cloud base, the frequency distribution of cloud base height in the manuscript may not be reasonable.

---

## Author Response (AR2)

**Response to Editor**

**Dear Editor & Prof.**

    We greatly thank you and the two reviewers for the thorough and valuable suggestions to our work. The manuscript has been polished and modified by professional organizations, and English has been greatly improved. We have made a point-to-point response to these opinions and suggestions, and believe that the quality of the manuscript has been promoted now. All comments have been modified and added in the revised manuscript (mark with blue font), and the responses to each comment are given below.

Thank you very much for considering our work!

Yours sincerely,

Yun Yuan and co-authors
Xi'an University of Technology
yunyuan_91@163.com
dihuige@xaut.edu.cn

**Response to anonymous Referee#1**

1. Lines 144 - 147 are not clear.

Therefore, $P_{new}(\lambda,r)$ is denoised by wavelet transform, threshold function is a soft threshold, wavelet base is sym7, and the number of decomposition layers is 5. Using wavelet function to reduce noise can avoid too much smoothing remove sharp signal changes due to clouds, and can also avoid the improper selection of moving average window.

**Answer:** Lines 144 – 147 are re-described as,

Therefore, wavelet denoising is used to deal with $P_{new}(\lambda,r)$, select symlets7 wavelet base as the wavelet decomposition basis function, the decomposition layer is 5, and the threshold value is the heursure based heuristic threshold value provided by MATLAB. Compared with the smooth function, wavelet denoising can avoid eliminating cloud signals with steep changes due to too much smoothing.

2. I think the units (%) for fig 16 in the revised manuscript are not correct, I believe these could be normalized units 0 to 1. I suggest the authors to check for proper units.

**Answer:** We have redefined the units of figures 16 and 18, and should not need the unit %. The Fig 16 has been changed in the manuscript.

3. Why the cloud cover shown in fig 17 is now high for December compared to the previous version?

**Answer:**

The cloud cover of the previous version was obtained by using the data of December 2021, and the cloud cover shown in fig 17 in the response and revised manuscript was obtained by using the data in December 2020. We have explained in the previous response document that the data from January to December 2021 be replaced by December 2020 to November 2021 (and divided into four seasons).

4. Line 426: the frequency of high-level ice clouds above 8 km is small. Is this due to the detection limit of the radar, if so please mention it?

**Answer:** The frequency of high-level ice clouds is small may be caused by the detection limited sensitivity of MMCR to small particles.

The corresponding content is added in lines 125-162 of the paper, as shown below,

L426-429: "Combined with the changing characteristics of cloud layers, it can be seen that during observation in Xi'an, the frequency of clouds below 3.5 km is the largest, and the frequency of high-level ice clouds or cirrus clouds above 8 km is small. " **and can be re-described as:**

L428-431: "Combined with the changing characteristics of cloud layers, it can be seen that during observation in Xi'an, the frequency of clouds below 3.5 km is the largest, and the frequency of high-level ice clouds or cirrus clouds above 8 km is small, which may be due to the limited detection sensitivity of MMCR at the top of high-level clouds where the particles size are very small."

5. Again, please check if the units are (%) for fig 18?

**Answer:** Figure 18 does not need the units (%), which has been changed in the manuscript.

**Response to anonymous Referee#2**

1. Line 64-65: "take" needs to be replaced with "mistake", "underestimated" should be "underestimation".

**Answer:**
"…but this method takes some real signals at the cloud bottom as noise and miss information at the cloud top, and resulting in overestimation and underestimated…"**and can be re-described as:**
"…but this method mistake some real signals at the cloud bottom as noise and miss some information at the cloud top, and resulting in overestimation and underestimation…"

2. Line 105: Please check "HT101", which is "TH101" in the response.

Answer: Sorry, it's a writing error in the response, and the "TH101" in the response should be "HT101".

3. Line 145: What is the full name of "sym" in "sym7"? It should be clarified.

**Answer:** Line 145: "sym" is the abbreviation of symlets, which is wavelet basis function, and has been described in the manuscript.

4. Line 153: Please add detailed description of the "baseline 1"and the "baseline 2" in the manuscript, and what is the difference between them?

**Answer:** Add description of baseline 1 and baseline 2 in lines 153-155, as follows,
Line 153-156: Get *Pnew-sp-smooth* after smoothing $P_{new\_sp}(\lambda, r)$. The slope $K_1$ of *baseline-1* obtained from the points (15, V1) and (endpoint, V2) on *Pnew-sp-smooth*, and *baseline-2* got by using $K_1$ and point (starting point, V0) as shown in Fig. 3b) and Fig. 4b).

5. Figure 3: Fig. 3c shows the vertical profile of SNR. However, "S" is marked on the horizontal axis. I'm confused whether it is the abbreviation of SNR and why the unit of S is "N".

**Answer:**
1) Fig. 3c) *SNR* in Shannon formula is the power ratio of signal to noise, which is a dimensionless unit. It has been explained in the response document, but Fig. 3c and Fig. 4c in the manuscript are flawed. Figures 3 and 4 are redrawn as follows.

2) Line 164, where $N$ is the pulse accumulation, and $P_{back}$ is the solar background noise power, **and can be re-described as:**
where $N$ is the pulse accumulation, $P_{back}$ is the solar background noise power, and *SNR* in Shannon formula is the power ratio of signal to noise, which is a dimensionless unit.

[Figure]

Fig. 3 Detection results of lidar at 12:13 on March 5, 2021. a) $P_{new\_sf}$ of the 1064 nm signal, b) $P_{new\_sp}$ of the 1064 nm signal, c) $SNR$ of $P_{new\_sf}$, d) cloud information detected

[Figure]

Fig. 4 Detection results of lidar at 22:44 on June 8, 2021. a) $P_{new\_sf}$ of the 1064 nm signal, b) $P_{new\_sp}$ of the 1064 nm signal, c) $SNR$ of $P_{new\_sf}$, d) cloud information detected

6. Line 184-185: "two parts" appears twice in this sentence.

**Answer:**

There are two parts in Fig. 6 includes two parts: recognition of cloud signals from Doppler spectra of MMCR and data quality control for MMCR, **and can be re-described as:**

Fig. 6 includes two parts: recognition of cloud signals from Doppler spectra of MMCR and data quality control for MMCR.

7. Line 203-205: Why you choose reflectivity of 20dBZ, velocity of 0.2m/s and spectra width of 0.3m/s as the thresholds?

**Answer:**

We use a large number data of MMCR in cloudless sky (ensure that it is the echo signals generated by planktonic) to analyze the numerical-frequency distribution characteristics of signals (reflectivity factor, radial velocity and velocity spectral width) in the range of 0 – 2km. We get that the reflectivity factor frequency is mainly distributed below -20 dBZ, the radial velocity is mainly distributed at -0.2m.s ~ +0.2m/s, and the velocity spectral width is mainly distributed above 0.3m/s. Therefore, we establish thresholds for eliminating signals of planktonic in the manuscript.

8. Line 214: There is no "Fig. 6b" in Figure 6, please check.

**Answer:**

The noncloud signals at the bottom (0–2 km) are effectively eliminated using the quality control algorithm shown in Fig. 6b), **and can be re-described as:**

The noncloud signals at the bottom (0–2 km) are effectively eliminated using the quality control algorithm shown in 2) of Fig. 6, and the accurate recognition of cloud boundary is realised in Fig. 7d).

9. Line 251: "The microwave radiometer accurately records the rainfall time" is mentioned, but the "similar to the following" is difficult to understand. The specific rainfall time recorded by the microwave radiometer should be illustrated.

**Answer:**

1) Line 251: at 06:00 CST (The microwave radiometer accurately records the rainfall time, similar to the following), **and can be re-described as:**

   at 06:00 CST (the microwave radiometer accurately records the rainfall time)

2) Line 279: the low-level cloud rained from 18:30 to 18:45 CST, **and can be re-described as:**

   the low-level cloud rained from 18:30 to 18:45 CST (the rainfall time is obtained by checking the microwave radiometer),

3) Line 332: When rainfall occurs (at 10:45 CST), **and can be re-described as:**

   When rainfall occurs (the microwave radiometer showed that rainfall occurred at 10:45 CST),

10. Line 258: The dotted line in Figure 8 is blue instead of "black". The color bars for a), b) and c) in Fig 8, Fig.10 and Fig.12 should be labeled with unit.

**Answer:**

The blue lines in Figs 8 and 10 have been changed to black lines. Corresponding units have been added in color bars of figures 8, 10 and 12. The units of color bar for a) in figures 8, 10 and 12 are dimensionless, and units of the color bars for b) and c) in figures 8, 10 and 12 are voltage / V.

[Figure]

Fig. 8 THI of the echo signal of the lidar @1064 nm from 08 to 09 June 2021. a) *SNR* of $P_{new\_sf}$, b) $P_{new\_sp}$ of the 1064 nm signal, c) cloud information detection results from lidar, d) reflectivity factor without quality control, e) reflectivity factor with quality control (black dotted line indicates rainfall time)

[Figure]

Fig. 10 THI of the echo signal of the lidar @1064 nm from 4 to 5 March, 2021. a) *SNR* of $P_{new\_sf}$, b) $P_{new\_sp}$ of the 1064 nm signal, c) cloud information detection results, d) reflectivity factor without quality control, e) reflectivity factor with quality control (black dotted line indicates rainfall time)

[Figure]

Fig. 12 THI of echo signal of the lidar and MMCR on 10 March, 2021. a) *SNR* of $P_{new\_sf}$, b) $P_{new\_sp}$ of the 1064 nm signal, c) cloud information detection results, d) reflectivity factor without quality control, e) reflectivity factor with quality control (black dotted line indicates rainfall time)

11. Figure 13: In the response version, the authors claim that "because there are a larger number of plotted points, the cloud bottom height around 0 km is appropriate", one should note that the bottom echo signal is not the height of the cloud base for precipitating cloud. If this height is treated as the cloud base, the frequency distribution of cloud base height in the manuscript may not be reasonable.

**Answer:** Sorry, we didn't clearly describe this problem in the response version. Our original intention is to express that in Figure 13, there is rainfall after 11:00 CST, and the reflectivity factor of MMCR has touched the ground. In this case, cloud bottom recorded by MMCR is 270 m instead of 0 m, and the cloud bottom is invalid according to the cloud bottom height recording guideline of case 5 in Table 3. Therefore, in Figs 16 and 18, we do not count the cloud bottom height of precipitating clouds measured by MMCR.